# SATA-BENCH: Select All That Apply Benchmark for Multiple Choice Questions

## Abstract

Current large language model (LLM) evaluations primarily focus on single-answer tasks, whereas many real-world applications require identifying multiple correct answers. This capability remains underexplored due to the lack of dedicated evaluation frameworks. We introduce SATA-BENCH, a benchmark for evaluating LLMs on Select All That Apply (SATA) questions spanning six domains, including reading comprehension, legal reasoning, and biomedicine. Our evaluation of 32 models demonstrates substantial limitations: the strongest model achieves only 75.3% Jaccard Index and 41.8% exact match accuracy. We identify three systematic biases underlying these failures: *(i) unselection bias:* models systematically avoid certain correct answer choices; *(ii) speculation bias:* models include incorrect answers when uncertain; and *(iii) count bias:* models consistently underpredict the number of correct answers.

## 1 Introduction

Large Language Models (LLMs) have demonstrated remarkable capabilities across diverse natural language processing tasks, with multiple-choice question answering becoming a standard evaluation framework [Pezeshkpour and Hruschka, 2024, Zheng et al., 2024]. However, current benchmarks assume a single correct answer per question, even though many applications require multiple valid responses, and because they rely on binary scoring that does not penalize speculation, they inadvertently encourage hallucination [**?**]. Consider content moderation systems that must flag posts

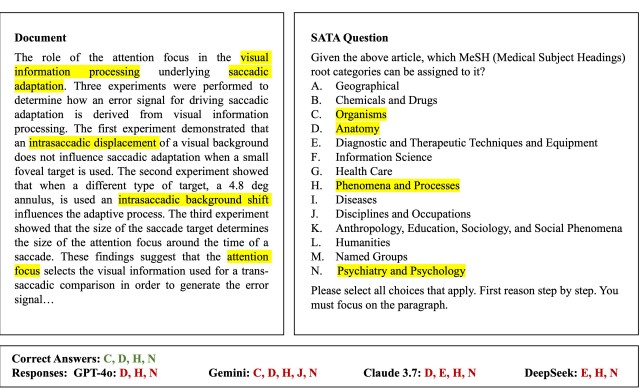

Figure 1: Representative example of an LLM failure on a SATA (Select All That Apply) question. Models often miss valid answers due to unselection, count, and speculation biases. Gemini speculates in this question while GPT-4o underselects. Other models may have unselection bias over C.

for several policy violations simultaneously, medical diagnosis tools that identify co-occurring conditions, or legal research platforms that classify documents under multiple relevant statutes. These scenarios represent Select All That Apply (SATA) tasks, where success depends not on choosing the single best option but on accurately identifying the complete set of correct answers. Despite their prevalence in real-world applications, SATA tasks remain underexplored in LLM evaluation, leaving a gap between benchmark performance and practical utility with direct implications for trustworthiness and safety. Existing evaluations overestimate model reliability by rewarding speculation, whereas SATA-specific metrics directly penalize speculative behavior.

To address this gap, we introduce SATA-BENCH, a comprehensive benchmark containing over 10,000 human-validated questions across six domains: reading comprehension, toxicity detection, news categorization, biomedicine, legal classification, and event analysis. Unlike existing multi-label classification datasets that often include dozens of possible labels and assume bag-of-words features, SATA-BENCH provides natural-language multiple-choice questions with 3–15 options and 2–10 correct answers, together with metrics that evaluate option-order effects, abstention behavior, and other phenomena unique to LLMs.

Our evaluation of 32 state-of-the-art models (including both proprietary LLMs and open-source alternatives) reveals substantial limitations in multi-answer reasoning. Even the best-performing model achieves only 41.8% exact match accuracy, missing the full correct set in nearly 60% of questions. Figure 1 illustrates a representative failure where models correctly identify some valid answers but systematically avoid others. We identify three systematic biases[1] underlying these failures: *unselection bias*, where models consistently avoid certain answer positions regardless of content; *count bias*, where models underestimate the total number of correct answers; and *speculation bias*, where models include incorrect options when uncertain rather than abstaining [**?**].

**Our Contributions.** The primary contributions of this paper are:

1. SATA-BENCH *Data Curation*: We curate a high-quality, diverse benchmark dataset explicitly designed to challenge LLMs on multi-answer tasks. SATA-BENCH contains more than 10K human-validated questions with multiple domains, varying difficulty levels, multiple correct answers, and carefully constructed distractors. In addition, we provide readability, confusion, and similarity analyses to ensure clarity, diversity, and task complexity across six domains.

2. *Comprehensive Evaluation*: We conduct the largest-to-date evaluation of 32 proprietary and open-source LLMs on SATA questions, revealing that even the strongest models achieve only 41.8% exact match accuracy and 75.3% Jaccard Index.

3. *Bias Diagnosis*: We identify and formalize *unselection*, *count*, and *speculation* biases as obstacles to solving SATA questions, and introduce multiple metrics to evaluate these biases.

## 2  SATA-BENCH Data Curation

Our objective is to develop a dataset that spans diverse tasks and domains while providing sufficient challenge to reveal differences in LLM capabilities. The curation process consists of three stages: (i) selecting source datasets, (ii) transforming them into SATA format, and (iii) filtering questions for readability, diversity, human validation, and clarity (see Figure **??**). We curated SATA-BENCH to include tasks in *Reading Comprehension* [Khashabi et al., 2018], *Text Classification* (News [Padmanabhan et al., 2016], Events [Event-Classification]), and *Domain Understanding* (Toxicity [Gehman et al., 2020], Biomedicine [PubMed-MeSH, 2021], Laws [Chalkidis et al., 2019]). Detailed dataset descriptions are provided in Appendix A.

## 3  Experiments

This section presents the experiments conducted to assess the capabilities of LLMs on SATA questions. The benchmark includes 16 proprietary models and 14 open-source models. (See Table 7 for full model cards.)

**Experimental Setup.** Because our benchmark contains diverse questions, we use a zero-shot evaluation. The system prompt specifies that each question has at least two correct answers, and we instructed the LLM to output the labeled result in JSONL format [Intelligence, 2024, Zhou et al., 2023]. Furthermore, we utilize a CoT prompting strategy as described in [OpenAI and el at, 2024]. We then extract the answer from the JSONL file using both exact match and fuzzy match. For cases where JSONL extraction fails (fewer than 3% of cases), we use Claude 3 Haiku and Human Labelers to extract the correct options from the answers provided. For smaller models, the percentage of cases where JSONL extraction fails exceeds 5%, making the above methods less reliable. Following [Hendrycks et al., 2021], we remove CoT and use the probability of the first output token to retrieve options. We hold out a dataset of 100 randomly sampled instances from the benchmark dataset to

---

[1]We use the term bias to highlight systematic tendencies in prediction (See Appendix **??** for mathematical definitions), not socioeconomic or demographic bias

Table 1: Compared to prior benchmarks [**?**], SATA-BENCH penalizes speculation, spans multiple domains, uses non-binary metrics, and includes multi-stage human annotations. Penalizing speculation means wrong answers receive lower scores than abstaining. Jaccard Index penalizes speculation: if ground truth is $A, B$ and model predicts $B, C$, $JI(JacardIndex) = 0.33$. if it does not speculate and predicts $B$, $JI = 0.5$. Thus, this scoring scheme gives a lower score to LLMs that speculate when uncertain.

| Benchmark | Scoring method | Binary grading | Penalizing speculation | Human labeling | # Domains |
|---|---|---|---|---|---|
| GPQA | Multiple-choice accuracy | Yes | None | Yes | 3 |
| MMLU-Pro | Multiple-choice accuracy | Yes | None | Yes | 57 |
| IFEval | Programmatic instruction verification | Yes[a] | None | No | 1 |
| Omni-MATH | Equivalence grading[*] | Yes | None | Yes | 1 |
| WildBench | LM-graded rubric[*] | No | Partial[c] | Partial | Varied |
| BBH | Multiple-choice / Exact Match | Yes | None | Yes | 23 |
| MATH | Equivalence grading[*] | Yes | None | Yes | 1 |
| MuSR | Multiple-choice accuracy | Yes | None | Yes | 1 |
| SWE-bench | Patch passes unit tests | Yes | None | No | 1 |
| HLE | Multiple-choice / equivalence grading[*] | Yes | None | Yes | 10+ |
| SATA-BENCH | Jaccard Index / Exact Match | Partial[b] | Yes | Yes | 6 |

[*] Grading is performed using language models, hence incorrect *bluffs* may occasionally be scored as correct.
[a] IFEval aggregates several binary rubric sub-scores into a composite score.
[b] Jaccard Index and Precision are not binary grading.
[c] Grading rubric (1-10 scale) may award hallucinated responses.

generate a threshold for each model with optimal Jaccard Index [Bogatinovski et al., 2022]. We select all options with a probability greater than that threshold value. Note that this method applies only to models with accessible token probabilities. We have also included the performance of non-expert humans on the benchmark (see Appendix G).

**Metrics.** We evaluate models using metrics across three categories: performance, selection bias, and count bias (details in Appendix H). Performance is measured using Exact Match (EM), Jaccard Index (JI), Mean Average Precision (Precision), and Mean Average Recall (Recall). Selection bias includes RStd [Zheng et al., 2024] and RSD [Croce et al., 2020, Reif and Schwartz, 2024]. We also introduce Selection Probability Divergence (SPD) to quantify unselection bias, a form of selection bias where models consistently avoid certain options. Count bias is assessed using the mean count difference (CtDif), mean absolute count difference (CtDifAbs), and count accuracy (CtAcc).

### 3.1 Key Observations

**SATA-BENCH is challenging and different.** 13 models achieve a JI above $70\%$, but none surpass $42\%$ EM. This shows that while models often identify some correct answers, they fail to consistently recover the full set.

Proprietary models generally achieve higher JI and Precision than open-source ones. Unlike other benchmarks, no single model dominates across all metrics. Notably, larger and more recent models do not always perform better. For instance, Claude 3 Sonnet outperforms Claude 3.5 Sonnet and Claude 3 Opus in exact match, though within the Claude family, larger models consistently have higher precision (e.g., Claude 3 Opus has the highest precision among the Claude 3 variants). According to [Anthropic, 2024, DeepSeek-AI and el at, 2024], these results contrast with performance on single-choice MCQ benchmarks such as MMLU [Hendrycks et al., 2021] and ARC [Clark et al., 2018], where larger or newer models typically show clear gains. Large reasoning models (LRMs) are slightly better than their non-reasoning counterparts in JI but failed to reduce selection and count bias. We provide a case study in Appendix **??** to investigate LRM's behavior.

**Models choose too few answers.** Nearly all LLMs tend to select fewer answers than required. For example, Llama 3.1 70B selects, on average, one fewer option per question than the correct number. Accordingly, it achieves the highest precision but the lowest Jaccard Index (JI). The tendency to under-select increases as the number of correct answers grows (Figure 9), which in turn depresses JI for questions with many correct choices (Figure 10). Even the best model achieves a CtAcc of only

Table 2: Performance comparison of 32 different LLMs across various metrics on SATA-BENCH. We highlight the best (**bold**) and second-best (underline) values. Columns labeled [(↑)] indicate higher-is-better; columns labeled [(↓)] indicate lower-is-better. Models with explicit reasoning capabilities are highlighted in *italic*. All numeric values are rounded to two decimal places. We retrieve exact labels for models evaluated using Inference-Based Retrieval + CoT prompting. For models evaluated under Probability-Based Retrieval, we select labels based on token probability thresholds.

| Model Name | Performance | | | | Selection Bias | | | Count Bias | | |
|---|---|---|---|---|---|---|---|---|---|---|
| | EM↑ | Precision↑ | Recall↑ | JI↑ | SPD↓ | RStd↓ | RSD↓ | CtDif | CtDifAbs↓ | CtAcc↑ |
| **Inference Based Retrieval + CoT** | | | | | | | | | | |
| *O3* | **41.77** | 87.50 | 81.22 | 73.91 | 0.38 | 6.79 | 0.06 | -0.39 | 0.94 | 46.12 |
| GPT4.1 | 40.49 | 85.52 | 85.66 | **75.23** | 0.13 | 5.98 | 0.06 | -0.04 | 0.85 | 45.52 |
| *GPT-OSS 120B* | 40.29 | 86.28 | 83.38 | 74.28 | 0.19 | 6.31 | 0.07 | -0.16 | 0.84 | 47.57 |
| *Grok 3 Think* | 39.71 | 83.93 | **86.31** | 74.40 | 0.30 | 6.26 | 0.07 | 0.06 | 0.93 | 44.24 |
| GPT4 | 39.47 | 85.90 | 83.17 | 74.11 | 0.21 | 6.63 | 0.06 | -0.20 | **0.82** | 46.61 |
| *Claude 3.7 Think* | 37.92 | 85.03 | 78.77 | 70.96 | 0.46 | 18.77 | 0.34 | -0.32 | 0.87 | 44.48 |
| Claude 3.7 | 37.82 | 85.35 | 77.15 | 70.98 | 0.49 | 6.59 | 0.25 | -0.43 | 0.93 | 43.58 |
| Claude 3 Sonnet | 36.49 | 84.58 | 78.81 | 70.72 | 0.36 | 7.37 | 0.07 | -0.35 | 0.83 | **48.00** |
| *Geimini 2.5 Think* | 36.46 | 84.58 | 83.25 | 72.58 | **0.12** | **4.76** | 0.06 | -0.01 | 0.88 | 43.76 |
| Claude 3.5 Haiku | 35.89 | 80.26 | 85.08 | 71.12 | 0.33 | 7.31 | 0.35 | 0.18 | 1.01 | 42.61 |
| Claude 3 Haiku | 35.64 | 83.59 | 80.16 | 70.63 | 0.42 | 6.24 | 0.07 | -0.22 | 0.85 | 47.15 |
| Claude 3 Opus | 35.59 | 86.97 | 77.19 | 70.15 | 0.62 | 8.26 | 0.07 | -0.52 | 0.93 | 44.36 |
| Gemini 2 Flash | 34.60 | 85.01 | 79.98 | 70.71 | 0.17 | 6.14 | 0.06 | -0.23 | 0.91 | 39.94 |
| GPT 4.1 mini | 33.46 | 86.05 | 78.23 | 69.90 | 0.30 | 6.69 | 0.06 | -0.39 | 0.97 | 38.61 |
| Nova Pro | 32.95 | 87.37 | 75.94 | 68.92 | 0.52 | 7.92 | 0.07 | -0.55 | 1.01 | 39.27 |
| Claude 3.5 Sonnet | 32.22 | 87.57 | 75.25 | 67.15 | 0.43 | 8.41 | 0.09 | -0.46 | 1.06 | 38.55 |
| Llama 3.1 405B | 30.17 | 86.24 | 75.31 | 67.18 | 0.33 | 6.90 | 0.45 | -0.39 | 1.02 | 36.30 |
| *Deepseek R1* | 28.17 | 84.62 | 72.36 | 64.49 | 0.94 | 17.44 | **0.03** | -0.57 | 1.13 | 33.52 |
| *GPT-OSS 20B* | 27.35 | 80.90 | 70.50 | 60.73 | 0.77 | 11.05 | 0.10 | -0.53 | 1.45 | 31.80 |
| Mistral Large V2 | 22.83 | 88.20 | 62.59 | 57.16 | 1.33 | 10.89 | 0.12 | -1.10 | 1.47 | 27.27 |
| Qwen Plus | 21.12 | 88.54 | 59.53 | 55.74 | 2.24 | 10.72 | 0.11 | -1.18 | 1.43 | 24.85 |
| Llama 3.2 90B | 18.30 | **89.56** | 60.80 | 55.78 | 1.84 | 11.10 | 0.27 | -1.09 | 1.41 | 24.30 |
| Llama 3.1 70B | 17.94 | **89.56** | 60.64 | 55.59 | 1.81 | 10.06 | 0.10 | -1.12 | 1.48 | 22.12 |
| Non-expert Human | 17.93 | 60.62 | 54.44 | 45.02 | 1.46 | 15.32 | 1.46 | -0.6 | 1.44 | 34.12 |
| **Probability Based Retrieval** | | | | | | | | | | |
| Mistral 8B | **14.73** | 81.46 | **53.23** | **46.63** | **11.42** | 19.47 | 1.27 | -1.35 | 1.95 | 21.01 |
| Llama3 8B | 13.82 | 80.30 | 47.37 | 43.64 | 12.09 | 17.85 | 1.09 | -1.59 | 1.88 | **22.00** |
| Bloomz 7B | 11.27 | 66.09 | 50.80 | 41.15 | 20.62 | 29.00 | 1.51 | **-0.87** | **1.71** | 20.09 |
| *DeepSeek R1 Distill 8B* | 8.85 | 72.20 | 45.81 | 40.02 | 13.38 | 21.62 | 1.14 | -1.29 | 1.75 | 20.42 |
| Qwen2.5 14B | 6.30 | 87.84 | 38.76 | 37.58 | 21.01 | 18.02 | **1.06** | -2.24 | 2.26 | 11.93 |
| Phi3 7B | 2.97 | **87.25** | 35.67 | 34.57 | 23.22 | 18.57 | 1.22 | -2.33 | 2.35 | 7.22 |
| *Phi4-mini-reasoning* | 2.12 | 77.98 | 30.82 | 29.69 | 21.62 | **13.90** | 1.59 | -2.37 | 2.39 | 7.35 |

48%, predicting the correct number of answers in fewer than half of the questions. We hypothesize that this behavior stems from models being primarily trained and evaluated on benchmarks with single correct answers, making them poorly suited for SATA tasks. A t-test confirms this under-selection: the mean of CtDif is significantly below 0, with $p = 1.70 \times 10^{-6}$.

**Models speculate a lot.** LLMs also over-select, consistently choosing incorrect options, with all models exceeding a 20% FPR. More than 70% of the models predict at least one incorrect choice more often than they produce exact matches, underscoring their speculating behavior. Interestingly, stronger-performing models tend to speculate more: hallucination rate and exact match are positively correlated ($r = 0.61$, $p = 8 \times 10^{-4}$). This dual trend suggests that as models improve in identifying correct answers, they also become more prone to speculation, highlighting the difficulty of disentangling genuine knowledge from overconfidence in LLM predictions.

**Unselection bias exists.** Some models exhibit a systematic tendency to avoid selecting certain labels, even when they are correct. When comparing Selection Probability Divergence (SPD) from our benchmark with 1,000 randomly simulated SPDs, Welch's t-test shows that LLMs' SPD is significantly higher than random ($p = 0.0467$). Even the best model in terms of selection bias (Gemini 2.5) underperforms on label M, with its recall rate 6.3% lower than its overall average recall (Figure 8).

## 3.2 Ablation Studies

We conducted ablation studies to test different strategies for improving model performance. We report the average results across three models selected for diverse profiles in terms of cost, open-source availability, and overall performance. The complete prompts are provided in Appendix J.3.

Table 3: Average performance of three models. The first column shows row numbers for reference.

| | Experiment | EM | Precision | RStd | CtDif |
|---|---|---|---|---|---|
| 1 | 1/2/3/4 | 35.50 | 82.99 | 10.22 | -0.37 |
| 2 | a/b/c/d | 30.69 | 83.10 | 11.56 | -0.26 |
| 3 | default | 33.00 | 84.62 | 7.37 | -0.25 |
| 4 | few shots | 28.35 | 76.61 | 17.33 | -0.42 |
| 5 | option by option | 30.50 | 86.28 | 4.81 | -0.64 |
| 6 | option few shots | 30.87 | 85.80 | 7.93 | -0.48 |
| 7 | with avg count | 27.33 | 76.17 | 14.90 | -0.40 |
| 8 | with count number | 53.95 | 83.30 | 3.45 | -0.08 |
| 9 | single choice | 45.53 | NA | NA | NA |

**Improving performance on SATA-BENCH is challenging.** We tried several approaches to improve performance, but none yielded consistent or significant improvements.

- **Changing the symbol** used for each answer choice did not improve the selection bias. We replaced the default option IDs from A/B/C/D to a/b/c/d and 1/2/3/4. While the 1/2/3/4 format achieved slightly better exact match accuracy, it also increased selection bias and reduced precision. Overall, we did not observe performance improvement by changing symbols (see rows 1-3 in Table 3).
- We provided **few-shot examples** in the prompt before the test models. However, this strategy did not lead to meaningful improvements in performance (see row 4 in Table 3).
- Inspired by survey science Smyth et al. [2006], Pew Research Center [2019], we instructed the models to **examine each option individually**. However, the models still selected too few options overall and did not improve performance (see rows 5-6 in Table 3).

Given more information, two approaches do improve performance and can provide additional insights into why the models struggle.

- **Providing the number of correct choices improves performance.** To understand how much error is due to the models' lack of knowledge regarding the number of correct options, we explicitly provided this information in the instruction for each question. This increased the exact match rate by 20.95 percentage points and reduced the selection bias metric RStd. However, when we instead provided the average number of correct choices across all questions in SATA-BENCH, performance declined (see rows 7-8 in Table 3).
- **Converting questions to multi-choice question with one correct answer.** For example, consider a question with three correct answers and six incorrect answers: we expanded it into three separate single-choice questions, each with one correct answer and six incorrect answers. We redefined the exact match rate as the percent of all original questions where a model answered all expanded questions correctly. This approach improved performance by 12.53% (see row 9 in Table 3), demonstrating that SATA questions are significantly harder for LLMs than single choice questions.

Both results suggest that while models can often identify individual correct answers, they lack awareness of how many correct answers exist, which contributes to their low performance.

# 4 Conclusion

We introduced SATA-BENCH, a dataset of over 10K human-validated SATA questions across six domains, and evaluated 32 LLMs. Even the best model achieves only 41.8% exact match accuracy, with failures driven by three systematic biases: unselection, count, and speculation. Although models can often identify individual correct options, our ablation studies show that they lack reliable mechanisms for estimating the correct number of answers. SATA-BENCH thus provides both a standardized benchmark and a diagnostic platform to analyze the LLM failure modes. We hope it will guide the development of models better suited for real-world applications where partial correctness is insufficient.

## Impact Statement

The introduction of SATA-BENCH marks a crucial advancement in evaluating Large Language Models (LLMs) on "Select All That Apply" (SATA) multiple-choice questions. By addressing a significant gap in existing benchmarks, which predominantly focus on single-answer multiple-choice tasks, SATA-BENCH challenges LLMs with real-world scenarios requiring multiple correct responses across domains such as reading comprehension, law, and biomedicine. This benchmark highlights the limitations of current LLMs, which struggle to accurately determine all valid answers, achieving a best-case exact match accuracy of only $41.8\%$.

SATA-BENCH 's impact extends beyond evaluation, as it reveals key biases in LLM decision-making, such as count bias and selection bias, which hinder performance on multi-answer tasks. To address these shortcomings, the development of the *Choice Funnel* algorithm demonstrates a novel approach to systematically improving LLM selection accuracy, significantly enhancing model performance in SATA tasks.

While the current focus is on knowledge-intensive domains, the potential for expansion into additional fields such as mathematics, coding, and instruction following is vast. SATA-BENCH can also be extended to free-text tasks, where the set of correct responses is not explicitly provided, and to other modalities, such as voice and vision. This would further refine LLM capabilities in handling complex, multi-faceted decision-making tasks. By pushing the boundaries of LLM evaluation, SATA-BENCH lays the foundation for the next generation of AI systems capable of more nuanced, flexible reasoning in diverse real-world applications.

## Limitations

**Memorization.** While we believe most questions in SATA-BENCH have not been seen during pretraining, we cannot fully rule out the possibility that some LLMs may have been exposed, even partially, to the source datasets. We do not have access to the pretraining data of proprietary models and therefore cannot conclusively assess memorization.

**Domain Coverage.** SATA-BENCH spans six diverse domains, including reading comprehension, biomedicine, and law. However, the total number of domains remains limited compared to larger-scale benchmarks such as MMLU, indicating that further expansion is needed for broader generalization.

**Text Modality.** Our benchmark is text-only. Real-world tasks often require multimodal reasoning (e.g., interpreting charts, images, or audio), which SATA-BENCH does not evaluate. We also do not address other data modalities, such as structured tabular data or sensor data, which are common in practical applications.

**Label Correctness.** Although we perform rigorous human validation and evaluations, we acknowledge that human beings can make mistakes. Some domains, such as biomedicine and law, are inherently complex and may contain subtle ambiguities. Thus, we cannot guarantee perfect correctness of all labels despite triple human annotation and agreement filtering.

**Language Limitation.** SATA-BENCH includes only English-language questions. Evaluating multilingual capabilities or cross-lingual transfer remains a work for the future.

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

# A   Dataset Description

In this section, we describe the original datasets and their characteristics in detail.

**Reading Comprehension** is a dataset of short paragraphs and multi-sentence questions that can be answered from the content of the paragraph. Some questions contain multiple correct answers. The dataset we use is from (https://cogcomp.seas.upenn.edu/multirc/). The metadata is licensed under the Research and Academic Use License.

We chose this dataset for the following 3 reasons.

1. The number of correct answer-options for each question is not pre-specified. This removes the over-reliance of current approaches on answer-options and forces them to decide on the correctness of each candidate answer independently of others. In other words, unlike previous work, the task here is not to simply identify the best answer-option, but to evaluate the correctness of each answer-option individually.

2. The correct answer(s) is not required to be a span in the text.

3. The paragraphs in our dataset have diverse provenance by being extracted from 7 different domains such as news, fiction, historical text etc., and hence are expected to be more diverse in their contents as compared to single-domain datasets. The goal of this dataset is to encourage the research community to explore approaches that can do more than sophisticated lexical-level matching.

**Toxicity** is adapted from RealToxicPrompts. The dataset select prompts from sentences in the OPEN-WEBTEXT CORPUS (Gokaslan and Cohen, 2019), a large corpus of English web text scraped from outbound URLs from Reddit, for which we extract TOXICITY scores with the PERSPECTIVE API. To obtain a stratified range of prompt toxicity, we sample 25K sentences from four equal-width toxicity ranges ([0,.25], ..., [.75,1]), for a total of 100K sentences. We then split sentences in half, yielding a prompt and a continuation, both of which we also score for toxicity. For each data point, we provide the definition for each category as well as shuffle the choices for each category. We only classify the case when the category's sum of prompt and continuation score is above 1.5 for each label. The dataset we use is from (https://huggingface.co/datasets/allenai/real-toxicity-prompts). The metadata is licensed under the Apache License.

**News** is processed from Reuters text categorization test collection dataset. It contains a collection of documents that appeared on Reuters newswire. There are originally 120 related topics, where each document can be related to multiple topics. There are two challenges related to this dataset preparation: 1. The number of topics can be too large for a small number of selections. 2. Some popular topics are commonly included in the documents, making a certain choice much more popular than other choices, which can bias the models in our study. With this in mind, we limit our selection to 10 options from the 120 topics for each documents, and the remaining choices are selected randomly from the topic pool; we also re-label the choices using unique mapping per document to keep the final answers evenly distributed between all letter choices (e.g. A/B/C/D...). The dataset we use is from (https://archive.ics.uci.edu/dataset/137/reuters+21578+text+categorization+collection). This dataset is licensed under a Creative Commons Attribution 4.0 International (CC BY 4.0) license.

**Biomedicine** is adapted from the PubMed MultiLabel Text Classification Dataset, which is a collection of research articles from the PubMed repository. Originally, these documents are manually annotated by Biomedical Experts with their Medical Subject Headings (MeSH) labels, and each article are described in terms of 10-15 MeSH labels. The adopted dataset has been processed and mapped to its root level with 15 distinct MeSH labels in total. The dataset we use is from (https://www.kaggle.com/datasets/owaiskhan9654/pubmed-multilabel-text-classification). This dataset is licensed under a CC0: Public Domain license.

**Laws** is adapted from EURLEX57K which contains 57k legislative documents in English from EUR-Lex (https://eur-lex.europa.eu) with an average length of 727 words. All the documents of the dataset have been annotated by the Publications Office of EU (https://publications.europa.eu/en) with multiple concepts from EUROVOC (http://eurovoc.europa.eu/). EURLEX contains 7201 concepts. There are two challenges when converting this dataset to multi-choice question answering dataset: 1. The 7201 concepts is too big a pool for a small number of selection, most documents have <10 concepts in this dataset. 2. Some popular concepts are included in a number of documents, making a certain choice much more frequent than other choices. This is problematic because it may force the

model to learn the popular letter of choice rather than the content of the questions. With this in mind, we limit our selection to 15 options from the 7201 topics pool for each document, and the remaining choices are selected randomly from the topic pool; we also shuffle and and re-label the choices using unique mapping per document to keep the final answers evenly distributed between each letter choice. The dataset we use is from (https://paperswithcode.com/dataset/eurlex57k). This dataset is licensed under Apache License.

**Events** is adapted from the "events classification biotech" dataset, which contains diverse biotech news articles consisting of various events. The curated dataset has 3140 questions with 5 choices of events for each document. Six choices are provided for each question. The dataset we use is from (https://paperswithcode.com/dataset/events-classification-biotech). This dataset is licensed under the Open Data Commons Attribution License (ODC-By) v1.0

## B  Dataset Filtering

The Biomedicine, Law, and Events datasets were originally multi-label classification tasks, which we adapted into SATA questions by creating distractor (incorrect) choices from the unselected labels. There are two challenges when converting these datasets to SATA format: 1. Many of them have a large label pool with only a few correct answers, which is not reasonable for multiple-choice questions. 2. There can be some popular answers frequently exist in the original data, making certain choices more frequent than others. This is problematic because it may force the model to learn the popular token of choice (e.g. Choice A/a/1) rather than the content of the questions. For example, the law dataset is originally from EUR-Lex data EUR-Lex [2018] contains 57k legislative documents in English (https://eur-lex.europa.eu) annotated by the Publications Office of EU (https://publications.europa.eu/en) with over 7k concepts from EUROVOC (http://eurovoc.europa.eu/). To address the first challenge, we kept an option-to-answer ratio between 2 and 3, considering the balance between the number of correct answers and incorrect choices. The distractors were sampled randomly from the topic pool. We also shuffled and and re-labeled the choices using unique mapping per question to keep the final answers evenly distributed between each choice token. An example question from each data source is shown in Figure 2.

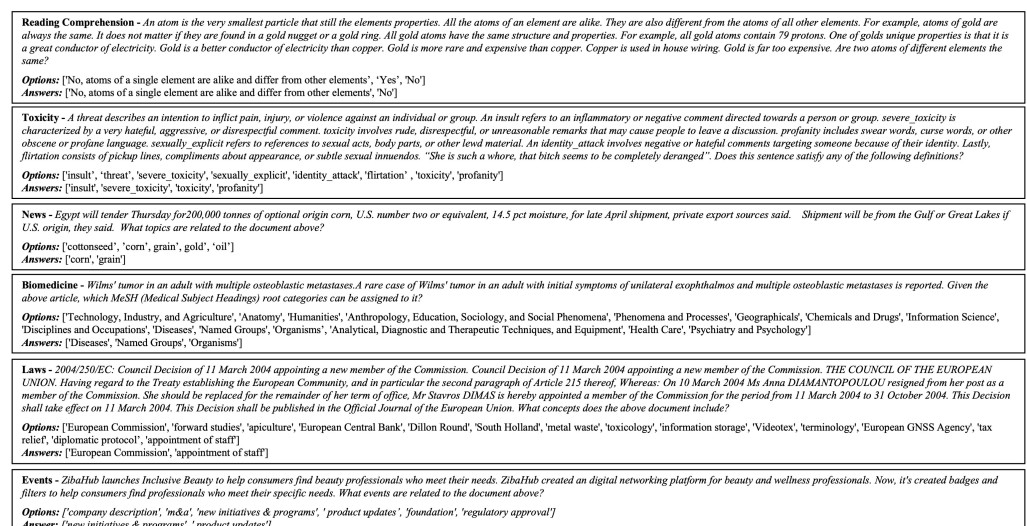

Figure 2: Representative examples of questions from various data sources used to construct SATA-BENCH.

### B.1  Initial Filtering

We manually filtered out questions that contain vague quantities, degrees of likelihood, temporal ambiguity, qualitative subjectivity, comparative uncertainty, general and undefined references. We use AWS Comprehend to remove questions that contain personal financial information or contact

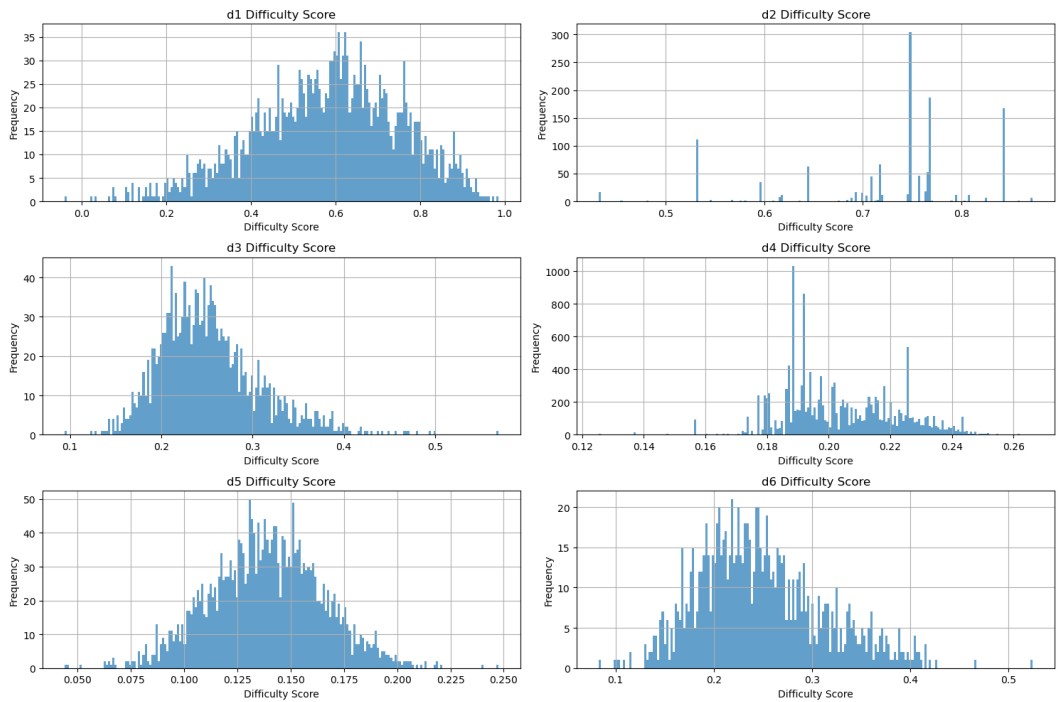

Figure 3: Confusion score distribution across all questions before filtering. d1: Reading Comprehension, d2: Toxicity, d3: News, d4: Biomedicine, d5: Laws, and d6: Events.

Table 4: Original data source statistics. We report the following metrics – n: number of instances, q: number of possible labels across the entire dataset, s: proportion of single-answer questions, m: mean number of correct answers, me: median number of correct answers, min: minimum number of correct answers, max: maximum number of correct answers, LC: label cardinality, r: ratio of the number of choices to the median number of correct answers (LC / me).

| Data Source | n | q | s | m | me | min | max | LC | r |
|---|---|---|---|---|---|---|---|---|---|
| Reading Comprehension | 5131 | na | 27% | 2.344 | 2 | 0 | 10 | 2-21 | na |
| Toxicity | 5994 | 8 | 60% | 2.639 | 2 | 2 | 7 | 8 | 4 |
| News | 11360 | 120 | 83% | 2.567 | 2 | 2 | 16 | 6 | 3 |
| Biomedicine | 50000 | 15 | 0.07% | 5.745 | 6 | 0 | 13 | 15 | 2.5 |
| Laws | 57000 | 7201 | 0.54% | 5.069 | 5 | 1 | 26 | 15 | 3 |
| Events | 3140 | 29 | 50.7% | 2.683 | 2 | 2 | 5 | 6 | 3 |

information. We leave questions that contain public available information such as the company name and address. All filtered words are mentioned below in Table 5.

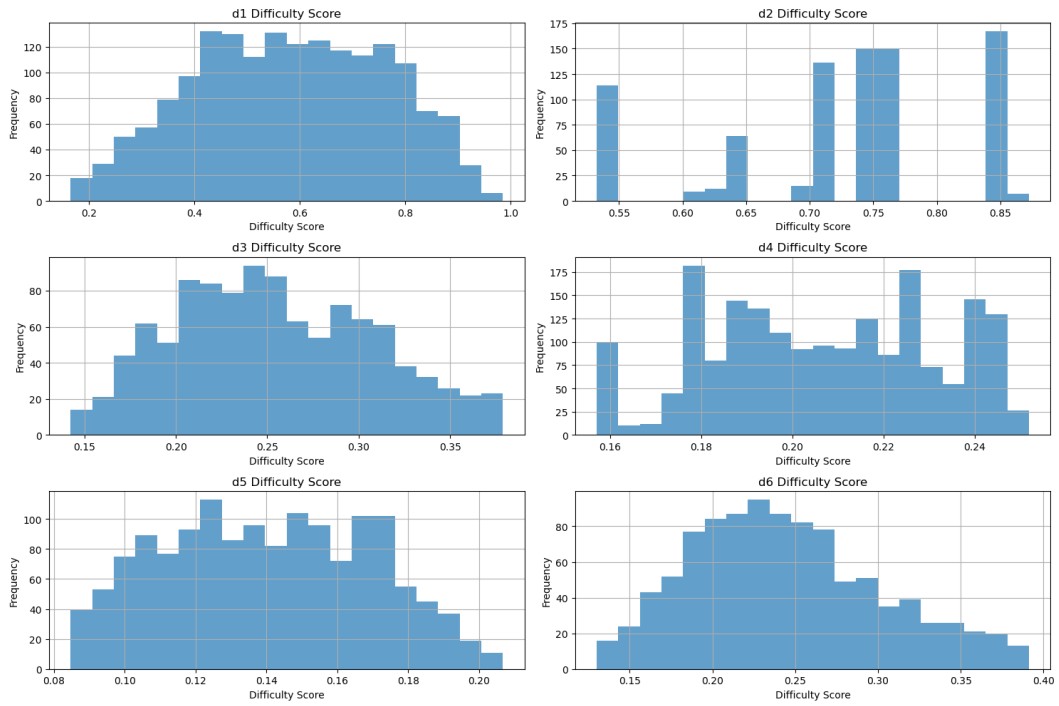

Figure 4: Confusion Score distribution of the filtered questions. d1: Reading Comprehension, d2: Toxicity, d3: News, d4: Biomedicine, d5: Laws, and d6: Events.

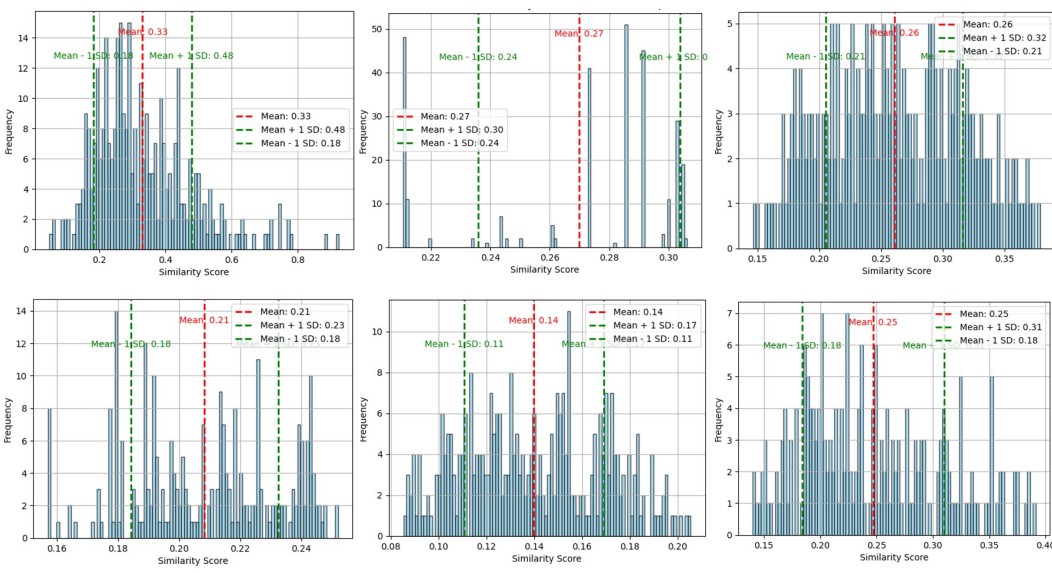

Figure 5: Confusion Score distribution separately visualized for each source dataset. (Left to right) Top row: Reading Comprehension, Toxicity, News; Bottom row: Biomedicine, Laws, Events.

Table 5: Identified categories of vague terms along with representative examples

| Category | Examples |
|---|---|
| Vague Quantities | some, several, many, few, a lot, plenty, numerous, various, partially, a handful, a bit, a portion |
| Degrees of Likelihood | maybe, possibly, probably, likely, unlikely, apparently, presumably, seemingly, conceivably, arguably, occasionally |
| Temporal Ambiguity | sometimes, often, rarely, occasionally, once in a while, from time to time, now and then, every so often |
| Qualitative Subjectivity | bad, nice, significant, substantial, important, interesting, sufficient, adequate, reasonable, moderate |
| Comparative Uncertainty | more or less, about, around, roughly, close to, kind of, sort of, nearly, almost, approximately |
| General and Undefined References | thing, things, anything, everything, whatever, such, kind, type, sort |

## B.2 Human Validation

Human validation is to ensure that the questions are unambiguous. Using humans to validate the question is inspired by [Tarrant et al., 2006, Moore et al., 2024]. For each question in the benchmark, we ask five annotators whether the question contains ambiguous information.

---

**Human Validation**

You are presented with the following:
Paragraph: *paragraph*
Question: *question*
Choices: *choice*
The question text and answer choices are clearly written:
*Strongly agree*
*Agree*
*Neither agree nor disagree*
*Disagree*
*Strongly Disagree*
Answers:

---

Once it is done, the total cost is tracked (1301.89), with 5 people per label at a cost of 0.012 each. We only select questions that are "Strongly agree" and "Agree" > 0.8.

## B.3 Human Labeling

To ensure that each question has a valid and correct answer, we conducted a comprehensive human evaluation. An initial manual inspection revealed that some questions lacked clearly correct answers. To verify answer correctness, we recruited three experienced annotators to review all 1,650 questions that remained after prior filtering and validation. Annotators were compensated at a rate of at least $35 per hour. Each question was independently evaluated by at least two annotators.

For each question, the original reference answer and four anonymized LLM-generated answers (from Claude 3.7, GPT-4 Turbo (O3), Grok 3, and Gemini 2.5) were provided. In cases where the two annotators disagreed, a third annotator reviewed the original answer, all LLM answers, and both annotators' decisions to determine the final label or to discard the question. Detailed annotation guidelines were provided below. As a result of this process, 47 questions were discarded due to ambiguity or disagreement, and an additional 46 were removed for quality-related issues.

## C Improving Performance on SATA Questions

The experimental results in Section 3 demonstrate that **Selection Bias** and **Count Bias** degrade LLM performance on SATA-BENCH, and that simple prompting strategies do not lead to significant improvements. This section focuses on improving performance on open-source models, which allows us to leverage token-level logits or probability estimates from the first token prediction.

To address **Selection Bias**, we draw from prior research on token debiasing methods [Choi et al., 2024, Zheng et al., 2024] in the MCQ setting, where selection bias is attributed to the *a priori* probability mass assigned by the model to specific option IDs. These methods propose various techniques to capture and remove such biases. We hypothesize that similar debiasing techniques can be adapted to mitigate unselection bias in SATA tasks. To address **Count Bias**,

---

**Algorithm 1:** Choice Funnel

**Input** : LLM $\pi_\theta$, SATA problem $\mathcal{T}$, option set $\mathcal{O}$, $NOTA$ stop option, $\tau$ confidence threshold

\# Initialize the selected option set
$\mathcal{R} \leftarrow \emptyset$
**while** $\mathcal{O} \neq \emptyset$ **do**
   \# Generate prompt with available options
   $\mathbf{P} \leftarrow \text{MakeSATAPrompt}(\mathcal{T}, \mathcal{O})$
   \# Get first token probability distribution and apply token debiasing
   $p \leftarrow \text{DebiasingFunction}(\pi_\theta(\cdot|\mathbf{P}))$
   \# Select option with highest probability
   $o \leftarrow \arg\max_{o \in \mathcal{O}} p(o)$
   \# 1. stop when "None of the above" is selected
   **if** $o = NOTA$ **then**
      | break
   **end**
   $\mathcal{R} \leftarrow \mathcal{R} \cup \{o\}$
   \# 2. stop when the confidence threshold is reached
   **if** $p(o) > \tau$ **then**
      | break
   **end**
   **if** $length(\mathcal{R}) = 1$ **then**
      | $\mathcal{O} \leftarrow \mathcal{O} \cup \{NOTA\}$
   **end**
   $\mathcal{O} \leftarrow \mathcal{O} \setminus \{o\}$
**end**
**Output** : $\mathcal{R}$

---

we retrieve the predicted probabilities of option IDs and select options whose probabilities exceed a predefined threshold. However, because SATA-BENCH includes a large option set, the probability distribution tends to decay rapidly, with most options receiving near-zero probability mass beyond the first few choices. This makes it challenging to establish a reliable threshold. Converting SATA questions into multiple binary classification problems helps but significantly increases inference cost.

**Choice Funnel Algorithm.** To improve model performance on SATA problems, we propose a decoding method called **Choice Funnel** (see Algorithm 1). This approach first adds an auxiliary option "None of the above". It then selects the option with the highest *first debiased token probability* and removes it from the option set. This process repeats iteratively until one of two stopping conditions is met: *(i) the model selects the "None of the above" option* or *(ii) the probability of the next selected option falls below a predefined confidence threshold.*

The idea of introducing "None of the above" (*NOTA*) comes from the traditional survey science domain, where options like "I don't know" (*idk*) are commonly included to improve the data collected in surveys [Schuman and Presser, 1996]. Recent research shows that survey design principles can inform LLM development [Eckman et al., 2024] and that LLMs exhibit similar biased response behaviors as humans Choi et al. [2024], Dominguez-Olmedo et al. [2024]. In our case *NOTA* outperforms *idk* (see ablation study in Appendix O.1).

The intuition behind the second stopping condition comes from our observation of model output probabilities, where the highest token probability tends to be lower at the beginning of iterations, since the model treats multiple options as equally correct. Later in the process, relatively higher probability is assigned to the final remaining correct option in the option set. We also show that Choice Funnel performs best when both stopping conditions are used together (see ablation study in Appendix O.3). Regarding the choice of *DebiasingFunction* in Algorithm 1, Choice Funnel is flexible and can incorporate any token debiasing method proven effective in MCQ settings. We demonstrate one such debiasing method in Section C. Additional ablation results on each sub-component of Choice Funnel are provided in Appendix O.2. Finally, the inference cost of Choice Funnel, measured by the number of model forward passes, scales linearly with the number of *correct labels* rather than the *total number of labels*. *This makes the method particularly efficient when the correct labels represent a small fraction of the option set.*

**Experimental Setup.** We adapted the PriDe algorithm [Zheng et al., 2024] as the token debiasing method in our experiments due to its label-free and computationally efficient implementation. It works by first estimating the model's prior bias towards specific option ID tokens (e.g., A, B, C) through random permutations of option contents in a small subset of test samples (10% in our experiments). We then use this estimated prior to adjust the prediction distribution on the remaining samples, separating the model's inherent positional and token biases from its task-specific predictions. Because the original PriDe algorithm was designed for standard single-answer MCQ settings, we modified it to better fit our SATA setting (see Appendix M).

We evaluate the performance of Choice Funnel against **three baseline methods** that rely on first-token probabilities: (i) First token probability with a fixed threshold, as defined in Section 3 (referred to as *first token*). (ii) Building on method 1, we apply PriDe debiasing method [Zheng et al., 2024] (referred to as *first token debiasing*). (iii) Convert each option into an individual binary yes/no question (referred to as *yes/no*). We expect *yes/no* to be a strong baseline, as it evaluates each choice independently. In this study, we use basic prompts (see Appendix J) and experiment with 7 LLMs from Table 2 that fall under the Probability Based Retrieval category (more details in Appendix N). For each model, we compute metrics reported in Table 2, and additionally report an *InfCost* metric to capture the number of model forward passes required for each method.

**Key Observations.** Choice Funnel consistently outperforms all three baselines across all 7 models in EM, SPD, and CtAcc (see Table 6). *Choice Funnel reduces unselection bias and count bias* – compared to the *first token* baseline, *Choice Funnel* achieves an average 56.16% reduction in *SPD* and 154.62% improvement in *CtAcc*, resulting in a 277.48% gain in Exact Match (EM) performance. While reasoning models also show improvements with Choice Funnel, we exclude these from aggregate calculations as their exceptionally low baselines would artificially inflate gains. When compared to our strongest baseline, the *yes/no* approach, *Choice Funnel* achieves a substantial

Table 6: Performance of various models on SATA-BENCH using different decoding methods. *Choice Funnel* achieves generally better performance, effectively reducing selection and count bias compared to three baseline methods. Best values in each column are highlighted in **bold**. Columns labeled [↑] indicate higher-is-better; columns labeled [↓] indicate lower-is-better. All numeric values are rounded to two decimal places.

| Model Name | EM↑ | Precision↑ | Recall↑ | JI↑ | SPD↓ | CtDifAbs↓ | CtAcc↑ | InfCost↓ |
|---|---|---|---|---|---|---|---|---|
| Mistral-8B + *first token* | 14.73 | 81.46 | 53.23 | 46.63 | 11.42 | 1.95 | 0.21 | **1650** |
| Mistral-8B + *first token debiasing* | 8.91 | 65.17 | 37.97 | 34.27 | 152.23 | 2.34 | 0.14 | 2534 |
| Mistral-8B + *yes/no* | 16.48 | 75.49 | **55.91** | 48.80 | 12.88 | 1.94 | 0.21 | 15517 |
| Mistral-8B + *choice funnel* | **20.24** | **86.03** | 55.78 | **52.56** | **8.50** | **1.74** | **0.27** | 4803 |
| Phi3-7B + *first token* | 2.97 | **87.25** | 35.67 | 34.57 | 23.22 | 2.35 | 0.07 | **1650** |
| Phi3-7B + *first token debiasing* | 1.76 | 67.92 | 28.24 | 27.47 | 175.24 | 2.50 | 0.05 | 2534 |
| Phi3-7B + *yes/no* | 25.45 | 78.41 | **72.40** | 60.03 | **1.39** | **1.64** | 0.30 | 15517 |
| Phi3-7B + *choice funnel* | **29.27** | 83.27 | 70.24 | **61.85** | 3.47 | 1.42 | **0.38** | 6339 |
| Qwen2.5-14B + *first token* | 6.30 | **87.84** | 38.76 | 37.58 | 21.01 | 2.26 | 0.12 | **1650** |
| Qwen2.5-14B + *first token debiasing* | 4.61 | 67.95 | 31.49 | 30.36 | 154.26 | 2.43 | 0.09 | 2534 |
| Qwen2.5-14B + *yes/no* | 25.64 | 79.80 | 60.56 | 56.18 | **2.76** | 1.52 | 0.31 | 15517 |
| Qwen2.5-14B + *choice funnel* | **27.82** | 85.69 | **67.07** | **61.12** | 3.80 | **1.42** | **0.35** | 6005 |
| Bloomz-7B + *first token* | 11.27 | 66.09 | 50.80 | 41.15 | 20.62 | **1.71** | 0.20 | **1650** |
| Bloomz-7B + *first token debiasing* | 7.09 | 59.07 | 38.41 | 32.05 | 149.17 | 2.19 | 0.15 | 2534 |
| Bloomz-7B + *yes/no* | 11.93 | 39.80 | 42.67 | 29.40 | 17.78 | 3.24 | 0.13 | 15517 |
| Bloomz-7B + *choice funnel* | **20.18** | **66.62** | **54.90** | **46.15** | **9.82** | **1.71** | **0.32** | 5440 |
| Llama3-8B + *first token* | 13.82 | **80.30** | 47.37 | 43.64 | 12.09 | 1.88 | 0.22 | **1650** |
| Llama3-8B + *first token debiasing* | 7.58 | 62.83 | 32.28 | 30.38 | 151.74 | 2.34 | 0.14 | 2534 |
| Llama3-8B + *yes/no* | 14.85 | 70.30 | **65.61** | 51.43 | **1.91** | 1.78 | 0.23 | 15517 |
| Llama3-8B + *choice funnel* | **19.88** | 78.69 | 56.19 | **50.36** | 7.75 | **1.66** | **0.33** | 4975 |
| Phi4-mini-reasoning + *first token* | 2.12 | **77.98** | 30.82 | 29.69 | 21.62 | 2.39 | 0.07 | **1650** |
| Phi4-mini-reasoning + *first token debiasing* | 1.27 | 59.77 | 25.74 | 24.51 | 156.16 | 2.32 | 0.07 | 2534 |
| Phi4-mini-reasoning + *yes/no* | 4.36 | 51.08 | **81.59** | 45.24 | 7.09 | 3.19 | 0.10 | 15517 |
| Phi4-mini-reasoning + *choice funnel* | **18.42** | 74.87 | 54.84 | **49.14** | **3.30** | **1.59** | **0.27** | 6003 |
| DeepSeek-R1-Distill-Llama-8B + *first token* | 8.85 | 72.20 | 45.81 | 40.02 | 13.38 | **1.75** | 0.20 | **1650** |
| DeepSeek-R1-Distill-Llama-8B + *first token debiasing* | 5.45 | 59.29 | 31.12 | 28.48 | 134.36 | 2.14 | 0.14 | 2534 |
| DeepSeek-R1-Distill-Llama-8B + *yes/no* | 0.12 | 40.31 | **89.51** | 40.19 | 27.96 | 5.73 | 0.01 | 15517 |
| DeepSeek-R1-Distill-Llama-8B + *choice funnel* | **14.36** | **75.56** | 45.56 | **42.87** | 12.37 | 1.87 | **0.21** | 4630 |

29.87% improvement in EM *while reducing model forward passes by 64.48% thanks to its early stopping mechanism*, demonstrating efficient inference scalability. Statistical significance testing (t-test) confirms that *Choice Funnel* significantly outperforms both *yes/no* and *first token debiasing* in EM and CtAcc, with a maximum p-value of 0.0079. While our models' parameter sizes (7B-14B) limit direct comparison to much larger proprietary models, Choice Funnel's performance on the *phi3-small* model still exceeds that of larger models such as Llama-90B and Mistral-Large V2 (see Table 2). This further underscores the effectiveness of our method. Additional ablation studies on the individual components of *Choice Funnel* are provided in Appendix O.

# D  Related Work

**SATA Benchmark.** Many existing MCQ benchmarks have only one correct answer and thus cannot test LLMs' ability to select multiple correct choices. On the one hand, existing SATA datasets, such as [Lewis et al., 2004, Kowsari et al., 2017, Aly et al., 2019, Katakis et al., 2008, Charte et al., 2015], have more than 30 labels per question to choose from. This makes it impractical for LLMs to identify all correct labels from such large label pools. Other SATA-style datasets test narrow, specialized capabilities, such as emotional understanding [Demszky et al., 2020] or music style understanding [Zhao et al., 2019], which are less emphasized in mainstream LLM benchmarks. Since most existing methods to solve SATA questions require converting questions to a bag-of-words [Liu et al., 2022], and as a result, most of the above datasets exist only in bag-of-words format, making them unsuitable for evaluating LLMs in our benchmark setting. To our knowledge, there is currently no existing LLM benchmark that consists exclusively of SATA questions.

**Selection Bias.** Many previous papers have discussed the tendency of LLMs to favor choices based on option order or specific symbols when answering MCQs [Gupta et al., 2024, Wei et al., 2024]. However, these papers have primarily focused on single-answer questions. A common approach to reducing selection bias involves calibrating output probabilities using the prior bias of an option ID [Zheng et al., 2024]. However, it remains unclear how to define or compute such priors in SATA questions.

# E  Hyperparameters

To ensure consistent and high-quality outputs across different models, we standardized the decoding hyperparameters for most model generations by setting the temperature to 0 (to promote deterministic outputs), top-p (nucleus sampling) to 0.95 (to allow for a balance between diversity and relevance), and a maximum token limit of 1,024 tokens. Recognizing the enhanced reasoning capabilities of certain models, we adjusted the configurations accordingly. For O3 and Grok 3, we set the thinking budget to be high. For Geimini 2.5 thinking and Claude 3.7 Thinking, we set the thinking budget to be 16k. For R1, we set max tokens 16k. This is to provide enough budget for reasoning models to finish thinking.

Table 7: Model cards summarizing specifications and details for all evaluated large language models.

| Model Name | Creator | Complete Model ID | Release | Hosting |
|---|---|---|---|---|
| O3 | OpenAI | o3-2025-04-16 | 04/16/25 | OpenAI API |
| GPT-4.1 | OpenAI | gpt-4.1-2025-04-14 | 04/14/25 | OpenAI API |
| Grok 3 Think | xAI | grok-3-mini-beta | 02/19/25 | xAI API |
| GPT-4-turbo | OpenAI | gpt-4o-2024-11-20 | 11/20/24 | OpenAI API |
| Claude-3.7 Sonnet Think | Anthropic | anthropic.claude-3-7-sonnet-thinking-20250219-v1:0 | 02/24/25 | AWS Bedrock |
| Claude-3.7 Sonnet | Anthropic | anthropic.claude-3-7-sonnet-20250219-v1:0 | 02/24/25 | AWS Bedrock |
| Claude-3 Sonnet | Anthropic | anthropic.claude-3-sonnet-20240229-v1:0 | 02/29/24 | AWS Bedrock |
| Gemini 2.5 Think | Google | gemini-2.5-pro-preview-03-25 | 03/25/25 | Vertex AI |
| Claude-3.5 Haiku | Anthropic | anthropic.claude-3-5-haiku-20241022-v1:0 | 10/22/24 | AWS Bedrock |
| Claude-3 Haiku | Anthropic | anthropic.claude-3-haiku-20240307-v1:0 | 03/07/24 | AWS Bedrock |
| Claude-3 Opus | Anthropic | anthropic.claude-3-opus-20240229-v1:0 | 02/29/24 | AWS Bedrock |
| Gemini 2 Flash | Google | gemini-2.0-flash | 02/05/25 | Vertex AI |
| GPT-4.1 mini | OpenAI | gpt-4.1-mini-2025-04-14 | 04/14/25 | OpenAI API |
| Claude-3.5 Sonnet | Anthropic | anthropic.claude-3-5-sonnet-20240620-v1:0 | 06/20/24 | AWS Bedrock |
| Llama 3.1 405B | Meta | meta.llama3-1-405b-instruct-v1:0 | 07/23/24 | AWS Bedrock |
| DeepSeek R1 | DeepSeek | deepseek.r1-v1:0 | 01/20/25 | AWS Bedrock |
| Mistral Large V2 | Mistral AI | mistral.mistral-large-2407-v1:0 | 07/24/24 | AWS Bedrock |
| Qwen Plus | Alibaba | qwen-plus-2025-04-28 | 04/28/25 | Alibaba API |
| Llama 3.2 90B | Meta | meta.llama3-2-90b-instruct-v1:0 | 09/25/24 | AWS Bedrock |
| Llama 3.1 70B | Meta | meta.llama3-1-70b-instruct-v1:0 | 07/23/24 | AWS Bedrock |
| Mistral 8B Instruct | Mistral AI | mistralai/Mistral-8B-Instruct-2410 | 10/09/24 | Hugging Face |
| Llama 3 8B | Meta | meta-llama/Llama-3.1-8B-Instruct | 07/23/24 | Hugging Face |
| BLOOMZ 7B | BigScience | bigscience/bloomz-7b1 | 07/11/22 | Hugging Face |
| DeepSeek R1 Distill 8B | DeepSeek | deepseek-ai/DeepSeek-R1-Distill-Llama-8B | 02/01/25 | Hugging Face |
| Qwen 2.5 14B | Alibaba | Qwen/Qwen2.5-14B | 09/19/24 | Hugging Face |
| Phi-3 7B | Microsoft | microsoft/phi-3-small-128k-instruct | 05/21/24 | Hugging Face |
| Phi-4-mini-reasoning | Microsoft | microsoft/phi-4-mini-reasoning | 04/15/25 | Hugging Face |

# F  Compute Resources

We use AWS Bedrock batch inference for large models' inference such as Claude3 Sonnet, Claude 3.5 Haiku, Claude 3 Haiku, Claude 3 Opus, Claude 3.5 Sonnet, Llama 3.1 405B, Mistral Large V2, Llama 3.2 90B, and Llama 3.1 70B. We use AWS cross-region inference for Claude3.7 Reason, Claude3.7, and Deepseek R1. We use official APIs from the respective providers for models such as OpenAI O3, GPT4.1, Grok3 Reason, GPT4, Geimini2.5 Reason, Gemini 2 Flash, GPT 4.1 mini, and Qwen Plus.

For experiments that require accessing model's hidden states and log probs. We run inference on one EC2 $p4d.24xlarge$ (Nvidia A100 40GiB GPU) instance and one EC2 $g5.4xlarge$ (Nvidia A10G 24GiB GPU) in Sydney(ap-southeast-2) region. We have also attached 8000GiB disk volume with AL2023 Linux OS image. We use HuggingFace and PyTorch as the main software frameworks.

# G  Non-expert Human Benchmark

To contextualise LLM results on SATA-BENCH, we recruited non-expert annotators on *Amazon Mechanical Turk*, adapting the instructions from  [Rein et al., 2023]. All 1604 questions was labelled as follows:

- **Task set-up.** Each question was presented with the original answer options *plus decoys* (e.g. ABCD→ ABCDEFGHIJK) to identify inattentive workers. Nine independent annotations were collected per item at a rate of *$0.84 per question*, matching the fair-wage recommendations of GPQA.

- **Quality safeguards.** Workers were: (i) informed that every item contains *at least two* correct answers; (ii) forbidden from consulting LLMs or other people, yet allowed to look up unfamiliar terms on Google/Wikipedia; (iii) required to spend $\geq 2$ *minutes* on each question. Submissions that selected any decoy, took $< 1$ min, or violated the lookup policy were discarded (7.1 %).

- **Label selection.** From the surviving pool, we randomly drew one annotation as the *human label*; single-choice answers were retained to keep the evaluation comparable to LLMs that sometimes return only one option.

|       | EM   | Precision | Recall | JI   | RStd | RSD  | SPD  | CtDif | CtAcc | CtDifAbs |
|-------|------|-----------|--------|------|------|------|------|-------|-------|----------|
| Human | 17.9 | 60.6      | 54.4   | 45.0 | 15.3 | 0.46 | 1.46 | −0.6  | 34.1  | 1.44     |

Table 8: Aggregate performance of crowd annotators on the SATA-Bench subset.

As anticipated, non-experts achieve modest exact-match and precision, yet their selection-bias metrics (RStd, RSD, SPD) resemble those of mid-tier LLMs. Crucially, they exhibit *smaller absolute count bias* (|CtDif|) and higher correct-count accuracy (CTACC), indicating superior intuition for the number of correct options even when individual labels are missed. These human baselines therefore offer a realistic point of comparison for evaluating LLM performance on specialised SATA tasks.

### G.1 Non-expert Human Benchmark Instructions

We have provided details on human benchmark instructions.

You will see a short **Paragraph**, a **Question**, and a list of answer options labelled `A B C D E F G H I J K L M N O`. Your task is to mark *all* choices that you believe are correct.

**Requirements and Expectations**

1. **External resources.** You may consult Google, Wikipedia, journals, textbooks, or any other online materials that help you understand the content. **Do not use AI assistants** (ChatGPT, Claude, Gemini, Grok, etc.) and do not ask other people.

2. **Effort.** Spend **at least 2 minutes** on each item before submitting. If you still feel unsure, keep researching until you are confident, or choose "*I don't know*" if you cannot answer reliably.

3. **Difficulty.** Many items are specialised and may require careful reading. Take your time; thorough work is valued more than speed.

4. **Feedback.** After answering, you may leave comments (e.g. ambiguity, unclear wording). Constructive feedback is highly appreciated.

**Suggestions and Strategies**

1. Look up definitions of every unfamiliar term in the paragraph, question, and answer options. Keep your notes open for quick reference.

2. Approach the question *independently*—do not try to guess a "majority" answer. Rely on primary sources (research articles, textbooks) whenever possible.

3. Remember that there are *at least two* correct letters, but possibly more. Select every option you deem correct.

**Fields Presented to You**

**Paragraph:** *{{paragraph}}*

**Question:** *{{question}}*

**Choices:** *{{A...O}}*

**Your Answers (mark all that apply):**

_______________________________________________

**Optional Feedback:** _______________________________________

# H   Metrics Definition

**Performance.** The standard SATA performance metrics [Tarekegn et al., 2024] are Exact Match (EM), Jaccard Index (JI), Mean Average Precision (Precision), and Mean Average Recall (Recall). EM captures a model's ability to select all correct answers without error, indicating its completeness in prediction. JI measures the fraction of predicted labels that match the ground-truth labels.

**Selection Bias.** Selection bias is a model's preference for selecting specific option IDs as answers, and is measured by RStd [Zheng et al., 2024] and RSD [Croce et al., 2020, Reif and Schwartz, 2024]. We also observe that some models prefer to avoid selecting certain options, which we call **unselection bias**. We introduce a metric called *Selection Probability Divergence* (SPD) to measure this type of bias. Appendix I gives more details on this issue and the design of SPD.

**Count Bias.** We observe that models tend to select fewer number of options compared to the ground truth. We refer to this as *Count Bias*. To evaluate the severity of this type of bias, we measure the following: (i) Mean difference between the number of selected options minus correct options (CtDif), (ii) Mean absolute difference between the number of selected options minus correct options (CtDifAbs), and (iii) Percentage of cases where the model selects exactly the correct number of options (CtAcc).

## H.1 Performance Metrics Definition

Here are some standard metrics used in the literature to track performance on SATA questions.

- **Exact Match** counts how many times the entire set of predicted labels for a sample exactly matches the entire set of ground truth labels. It is then divided by the total number of samples. A perfect exact match score (1.0) means the model got every instance's labels exactly correct.

- **Jaccard Index** calculates the fraction of predicted labels that exactly match the ground truth labels—or put differently, divide the size of the intersection of predicted and true labels by the size of the union of predicted and true labels, and then average this ratio across all instances for the final score. This metric treats each label decision independently and is a good measure when we care about partial correctness in multi-label settings.

- **Recall** looks at how many labels were correctly predicted (intersection) out of how many total true labels exist. Then it averages this fraction across all instances.

- **Precision** calculates how many labels were correctly predicted (intersection) out of all the labels the model predicted. Then it averages this fraction across all instances.

## H.2 Selection Bias Metrics Definition

Here are some standard metrics to track SATA questions selection bias. These metrics are extension of existing selection bias literature.

- **Standard Deviation of Recalls (RStd)** is the standard deviation of the class-wise recall:

$$\text{RStd} = \sqrt{\frac{1}{k} \sum_{i=1}^{k} (r_i - \bar{r})^2},\tag{1}$$

  where $k$ is the number of choices, $r_i$ is the recall of the $i$-th class, and $\bar{r}$ is the arithmetic mean of $r_i$ values. Note that our recalls are calculated at the label level since this is multi-class question [Zheng et al., 2024]

- **Relative Standard Deviation (RSD)** is the class-wise accuracy standard deviation normalized by the overall accuracy:

$$\text{RSD} = \frac{\sqrt{\frac{1}{k} \sum_{i=1}^{k} (s_i - \bar{s})^2}}{\bar{s}},\tag{2}$$

  where $k$ is the number of choices, $s_i$ is the accuracy of the $i$-th class, and $\bar{s}$ is the mean accuracy averaged across classes. Please note that our recalls are calculated at the label level since this is multi-class questions [Croce et al., 2020, Reif and Schwartz, 2024]

## H.3 Count Bias Metrics Definition

- **CtDif** calculates the average difference in count between predicted and actual selected options. A positive value indicates that the predictions tend to select more options than the actual answers, while a negative value suggests the opposite.

- **CtDifAbs** calculates the absolute value of the average difference in count between predicted and actual selected options. A larger value indicates that the predictions tend to select the number of options that are different from the correct number of options.

- **CtAcc** calculates the proportion of predictions that select the exact same options as the ground truth labels. It provides a measure of how often the model selects the same number of answers as the true answer set.

## H.4 Additional Metrics Definition

- **InfCost** measures the number of model forward passes used for a method to complete the benchmark. A larger value indicates that the method requires more compute FLOPs and is thus more expensive. A small value indicates the method requires fewer compute FLOPs and is thus more cost-effective.

# I  Unselection bias metric

We view a SATA problem as multiple binary selection problems, where each option is examined independently to be selected or passed. In our experiments, we have observed that LLMs tend not to select (i.e., skip) certain labels more frequently than others. To quantify this non-selection bias, we define a metric below, named selection probability divergence (SPD), to measure the misalignment between the ground truth and the LLM's prediction.

$$\text{SPD} = \sum_{i=1}^{k} \left(1 - \frac{q_i}{p_i}\right) \ln \frac{p_i}{q_i}, \tag{3}$$

where $k$ is the number of choices, $p_i$ is the ground truth probability of label $i$ being one of the correct choices, and $q_i$ is the prediction probability of label $i$ being one of the selected choices.

SPD has a minimal value of 0 at $q_i = p_i$ for all $i$, when the prediction aligns with the ground truth. SPD diverges as $q_i \to 0$ while $p_i$ is finite for any $i$, when the LLM shows a non-selection bias against a particular label. SPD also diverges as $p_i \to 0$ while $q_i$ is finite for any $i$, when the LLM shows a selection bias toward a particular label. In this sense, SPD serves as a metric to measure the disagreement of choice probability between the ground truth and the prediction, reflecting both under-selection and over-selection. (See Appendix I.2 for the mathematical analysis.)

## I.1  Behavior of SPD Metric

We conduct a numerical experiment to compute SPD with varying $p_i$ and $q_i$. We set the number of choices to 4, and use a Boolean list of size 4 to indicate which options are correct. Eg. for choices A, B, C, and D, the list [True, False, True, True] means the answer to the SATA question is ACD.

For the ground truth list, we sample each element of the Boolean list with a ground truth probability, p. For the prediction list, we sample the first element of the Boolean list with a prediction probability, q, and sample the other elements with probability p. With this setting, we focus on the misalignment between the ground truth and the prediction in a single label (the first label in this case).

We repeat the above sampling process $M$ times, and compute the True rate of each option for the ground truth $p_i$ and the prediction $q_i$, with $i = 1, 2, 3, 4$. We then substitute the numbers into Eq. (3) to calculate SPD. Note that in the current setting, $p_i = \text{p}, \forall i$, and $q_1 = \text{q}$, $q_{2,3,4} = \text{p}$.

Figure 6 shows the SPD-q curves under different values of the ground truth probability p. Each curve is obtained by averaging over 100 replicates, and the shaded area shows the standard deviation. The minimal value of SPD is 0 and occurs at q = p.

## I.2  Sensitivity of SPD to label probability ratio

We analyze the behavior of SPD as the relationship between $p_i$ and $q_i$ changes. We first define the ratio of the two probabilities as $r_i \equiv q_i/p_i$, $i = 1, 2, \ldots, k$, and rewrite the SPD definition Eq. (3) as

$$\text{SPD} = \sum_{i=1}^{k} (1 - r_i) \ln \frac{1}{r_i}. \tag{4}$$

As the misalignment between the ground truth and the prediction grows, either with $r_i \to 0$ or $r_i \to +\infty$, SPD diverges according to Eq. (4). Therefore, a large value of SPD reflects the disagreement of the choice probability between the ground truth and the prediction.

To find the minimum of SPD, we take the partial derivative with respect to each variable $r_i$, and set it to be 0. Then we have the equations below.

$$\frac{\partial \text{SPD}}{\partial r_i} = \ln r_i + \frac{r_i - 1}{r_i} = 0, \quad i = 1, 2, \ldots, k. \tag{5}$$

This set of equations has only one real solution:

$$r_i = 1, \quad i = 0, 1, \ldots, k. \tag{6}$$

Thus the SPD is minimized when $q_i = p_i$, *i.e.* when the prediction probability matches the ground truth probability for each option and when there is no bias toward or against any choice. The minimal value of SPD is 0.

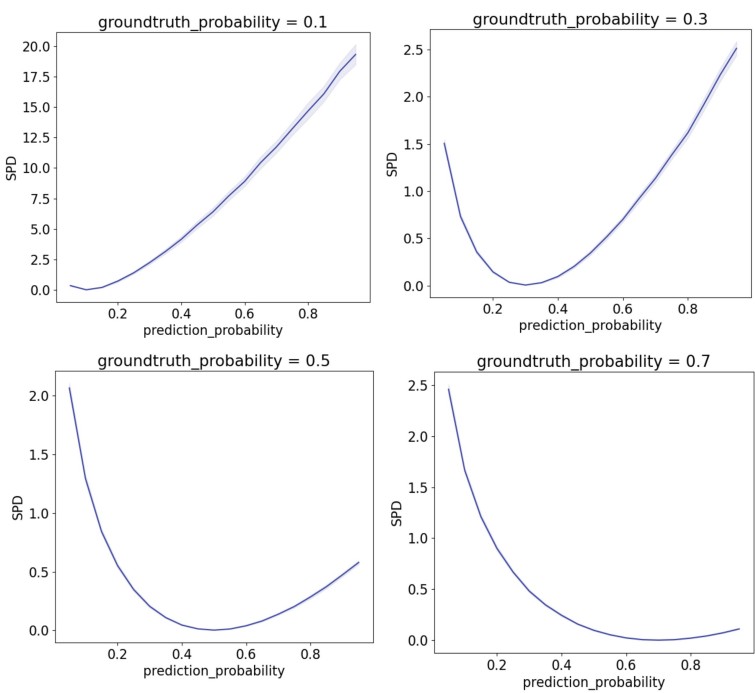

Figure 6: Relationship between Selection Probability Divergence (SPD) and prediction probability (q) across different ground truth probabilities (p). The curves are averaged over 100 replicates, and the shaded area represents the standard deviation. In each plot, the minimal value of SPD is 0 at q = p, when the prediction aligns with the ground truth.

## J   Prompts used in experimentation

### J.1   Prompts for open-source models

We designed simple, basic prompts without elaborate prompt engineering for all experiments with open-source models in Section 3. The main reason is that we want to avoid potential biases introduced by complex prompt engineering, thereby emphasizing the evaluation of the method itself.

### J.1.1   Choice funnel Prompt

This prompt is used for *Choice Funnel* as well as two baseline methods: *first token* and *first token debiasing*

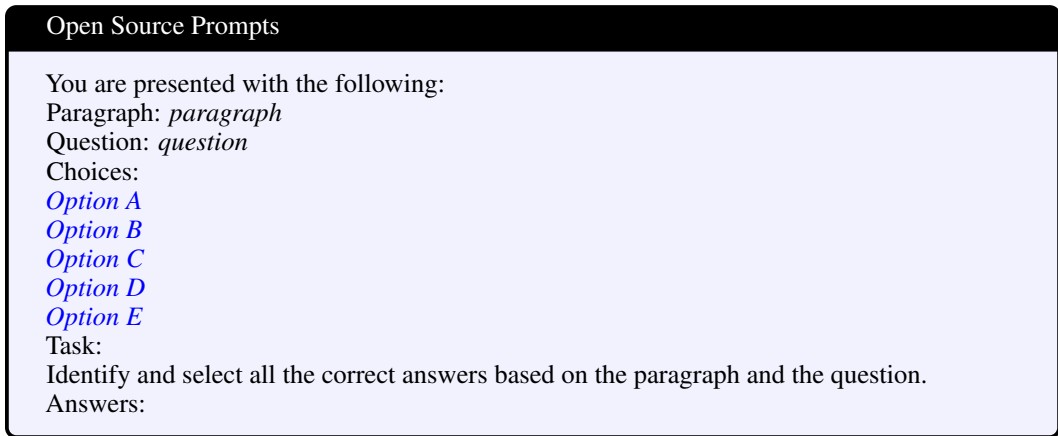

### J.1.2 Yes/No for open-sourced models

This prompt is used for *yes/no* baseline method to compare against *Choice Funnel*.

---

**Yes/No Prompts**

You are presented with the following:
Paragraph: *paragraph*
Question: *question*
Statement: *Option A |B |C |D |E*
Task:
Determine if the statement answers the question correctly and reply with "Yes" or "No" only.
Answer:

---

## J.2 Prompts for proprietry model

### J.2.1 Problems for current MCQ prompts

Existing benchmarks [Intelligence, 2024] use the following prompts for MCQ questions and then use exact match to get the correct option.

---

**Example MCQ Prompt 1**

What is the correct answer to this question: *question*
Choices: *choices*.
Let's think step by step:
Based on the above, what is the single, most likely answer choice?
Answer in the format:
correct answer is (insert answer here).

---

**Example MCQ Prompt 2**

Given the following question and four candidate answers (A, B, C, and D), choose the best answer.
Question: *question*
Your response should end with:
The best answer is [the_answer_letter]
where the [the_answer_letter] is one of A, B, C, or D.

---

However, it does not work well when there are multiple answers as LLMs are likely to repeat the content of the option before output the second choice. As an example, the below answer is A. Wheat, C. Grain instead of AC, which makes it hard to parse.

> **Example MCQ Prompt Example 1**
>
> What is the correct answer to this question: Bangladesh passed on its weekend tender for 100,000 tones of optional origin soft wheat, private export sources said. What topics are related to the document above?
> Choices: A. Wheat, B. Barley, C. Grain, D. Retail, E. Money-supply, F. Sugar]. Let's think step by step:
> Based on the above, what are all most likely answer choice?
> Answer in the format:
> correct answer is (insert answer here).
> Answer (from GPT4o): Let's analyze step by step:
> The document mentions wheat: The phrase "100,000 tonnes of optional origin soft wheat" directly refers to wheat.
> Relevant choice: A. wheat. The document does not mention barley: There is no reference to barley anywhere in the text.
> Not a relevant choice: B. barley. Grain is a broader category that includes wheat: Since wheat is a type of grain, it makes sense to include grain as a related topic.
> Relevant choice: C. grain. The document does not mention retail: The context of the tender does not relate to retail or consumer sales.
> Not a relevant choice: D. retail. The document does not mention money supply: There is no financial or monetary context provided.
> Not a relevant choice: E. money-supply. The document does not mention sugar: There is no reference to sugar in the text.
> Not a relevant choice: F. sugar. Final Answer:
> Correct answer is **A. Wheat, C. Grain**.

> **Example MCQ Prompt Example 2**
>
> What is the correct answer to this question: Bangladesh passed on its weekend tender for 100,000 tones of optional origin soft wheat, private export sources said. What topics are related to the document above? Choices: A. Wheat, B. Barley, C. Grain, D. Retail, E. Money-supply, F. Sugar. Let's think step by step:
> Based on the above, what are most likely answer choices?
> Answer in the format:
> correct answer is (insert answer here).
> Answer (from GPT4o): Let's analyze the question step by step:
> Key Subject of the Document: The document discusses Bangladesh passing on its tender for 100,000 tonnes of optional origin soft wheat.
> Relevant Topics:
> The keyword wheat clearly relates to the document. Wheat falls under the broader category of grain, making it a relevant choice. Other terms like barley, retail, money-supply, and sugar are not directly related to the subject of the document. Most Likely Answer Choices:
> A. Wheat: Directly mentioned in the document. C. Grain: Relevant as wheat is a type of grain. Correct Answer:
> The correct answer is **A Wheat, C Grain**.

We can observe that parsing the above prompts is much harder because we do not know when they will output the option ID plus answers, and we cannot blindly extract all matched option IDs. Following [Intelligence, 2024] page 38, we used the JSON format to extract answers. Closed-source LLMs have shown good instruction following capability for JSON format output [Zhou et al., 2023]. Also, python has many packages that can do fuzzy JSON matching, which can be used to handle edge cases. We extract as many JSON files as possible from each output and combine the extracted answers together as the final answer to the question.

One can observe that our proposed prompts can easily extract the answer because they contain only the option IDs.

### J.3 Ablation Prompts

### J.3.1 Few Shot prompt

We report few few-shot prompt where the number of examples is equal to 5.

> **Few Shots Prompt**
>
> Given the following question and four candidate answers (A, B, C, and D), choose the best answer.
> Question 1: *question 1*
> Option 1: *option 1*
> Answer 1:*correct option json 1*
> Question 2: *question 2*
> Option 2: *option 2*
> Answer 2: *corect option json2*
> ...
> Question 5: *question 5*
> Option 5: *option 5*
> Answer 5:*correct option json 5*
> Question: *question*
> Option: *option*
> Please select all choices that apply. You must focus on the question and select all choices that apply. Let's think step by step: You must present your selected option IDs in the following JSON format: $\{"choices" :< A|B|C|D|E|F|G|H|I|J|K|L|M|N|O >\}$

## J.4 Think Option by Option prompt

Inspired by Smyth et al. [2006], Pew Research Center [2019], we instruct LLM to understand each options and analyze each answer independently.

> **Choice-by-choice Prompt**
>
> Given the following question and four candidate answers (A, B, C, and D), choose the best answer.
> Question: *question*
> Option: *option*
> Let's think through this step by step:
> 1. First, let's understand what the question is asking...
> 2. Now, let's evaluate each option individually...
> 3. Therefore, the correct answers are...
> You must present your selected option IDs in the following JSON format:
> $\{"choices" :< A|B|C|D|E|F|G|H|I|J|K|L|M|N|O >\}$

## J.4.1 Few Shot Option prompt

We further provide a few examples to teach LLMs how to think option by option, but it still does not improve the performance.

### J.4.2 Prompt with Average Options Count

### J.4.3 Prompt with Correct Number of Options

### J.4.4 Single Choice Prompt

To ensure consistency, we use a similar prompt for single choice. We use the same method to retrieve the correct choices. If there is more than one correct choice, we randomly sample from among them.

**Single Choice Prompt**

Given the following question where there is only one correct answers, choose the correct answer.
Question: *question*
Choices: *choices*
Please the correct choice that apply.
Let's think step by step: You must present your selected option IDs in the following JSON format: {"*choice*" :< $A|B|C|D|E|F|G|H|I|J|K|L|M|N|O$ >}

## J.5 Prompt with Numeric Option

For numeric options, it is hard to retrieve since the number of options can be above 10, and the previous retrieving method could retrieve 12 as 1 and 2. We instruct LLMs to produce correct answers in ascending order. We start by retrieving a larger number that is above 10. For each successful retrieval, remove that number from the output. This way, we can avoid the above scenario.

**Numeric Prompt**

Given the following question where there is more than one correct answer, choose all correct answers.
Question: *question*
Choices: *choices*
Please select all choices that apply. You must focus on the question and select all choices that apply. You must present your answers in ascending orders. Let's think step by step: You must present your selected option IDs in the following JSON format: {"*choices*" :< $1|2|3|4|5|6|7|8|9|10|11|12|13|14|15$ >}

## J.6 Prompt with small alphabet Option

**Small Alphabet Prompt**

Given the following question where there is more than one correct answer, choose all correct answers.
Question: *question*
Choices: *choices*
Please select all choices that apply. You must focus on the question and select all choices that apply. Let's think step by step: You must present your selected option IDs in the following JSON format: {"*choices*" :< $a|b|c|d|e|f|g|h|i|j|k|l|m|n|o$ >}

# K   Inference Error Handling

For $2.897\%$ of all cases, we cannot find any match in JSON format, so we use Claude 3 Haiku to extract the final labels. To be specific, we adopt the following system prompt:

For all cases below, our Claude 3 haiku is able to accurately produce the correct outcome.

Table 9: Comparison of raw LLM outputs and the extracted labeled results obtained using Claude 3 Haiku.

| LLM Output | Claude 3 Haiku Extraction |
|---|---|
| I can't fulfill that request. | NaN |
| `"choices": { "choice": "B" }` | B |
| `{{ "choice": <B \| E \| H \| J \| L \| M \| O> }} }}'''json` | BEHJLMO |
| `{ "choice": [] } '''` | NaN |

We then use Amazon Groundtruth labeling to check whether Claude 3 Haiku correctly parses the answer. Of those, only 47 cases were labeled as No or Yes with confidence lower than 0.6. We manually investigated those 47 cases and found that only four were actually incorrect.

Table 10: Examples of LLM outputs and corresponding extraction results where Claude 3 Haiku produced incorrect extractions.

| LLM Output | Claude 3 Haiku Extraction | Human Corrected Answers |
|---|---|---|
| Letś analyze the text and MeSH categories step by step:...: your selected option IDs - C (Organisms), your selected option IDs - E (Phenomena and Processes), your selected option IDs - G (Chemicals and Drugs) | CE | CEG |
| `{{ "choice": <D \| E \| K \| L \| M> }} }}'''json` | DELM | DEKLM |
| `{ "choice": "choice": "N"oneyour se-lected option IDs } '''` | N | NaN |
| Letś analyze the document step by step: ... your selected option IDs your selected option IDs. Based on this analysis, the applicable choices are A, B, C, and E. | ABC | ABCE |

# L   More Details on Key Observations

**Unselection Bias.** FP/FN means False Positive Count divided by False Negative Count. If a model has 100 False Negative cases of A, it means that the model has not predicted A in 100 cases where it should have predicted A. If a model has 20 False Positive cases of A, it means that the model has predicted A in 20 cases where it should not have. The low FP/FN rate means that out of all cases, the model tends not to predict A instead of overpredicting A. Due to Count Bias, most of the models have FP/FN rate below 1. However, almost all models has one label with an extremely low FP/FN

rate. For example, Claude3-Haiku has a label A FP/FN rate equal to 0.27 while its second worst is 0.48 as shown in Figure 8.

Recall Difference is another metric to demonstrate unselection bias. Low recall on certain label means that LLMs' incapability of predicting certain labels correctly. As shown in Figure 7, there are many models whose worst label is more than $5\%$ below their average performance.

**Count Bias.** Figure 9 shows that nearly all models select too few responses and that this tendency increases as the number of correct answers increases. Figure 10 shows that EM also decreases as the number of correct answers increases. This shows that LLMs tend to underpredict the number of correct choices.

## M  PriDe Debiasing Algorithm Adaptation for SATA

### M.1  PriDe Introduction

The original PriDe algorithm [Zheng et al., 2024] is designed for processing MCQ question sets with fixed option set length (usually 4). It works by observing the probability changes when performing permutations of option IDs for each question, and it can compute *priors*, which is known as the probabilistic mass that the model a priori assigns to option ID tokens.

Here is an example to better illustrate the process:
Given a question set with 4 options, we compute the prior of each question from 10% of the data, take the average on each option ID position and then we get:
$$P(prior) = [0.4, 0.2, 0.2, 0.2]$$

The list corresponds to probabilities for ABCD. In this case we can see that the model biases towards option "A". Now given a new question with probabilities computed as:
$$P(observed) = [0.5, 0.3, 0.1, 0.1]$$

Without debiasing model will select option "A" as top answer. We need to subtract prior:
$$P(debiased) = P(observed) / P(prior)$$
$$P(debiased) = [1.25, 1.5, 1.0, 0.5]$$

Option "B" becomes top-1 after we remove the heavy prior on "A". To learn more low-level details, please refer to the original paper [Zheng et al., 2024]).

### M.2  Limitation of Original Algorithm.

However, the prior is computed on a fixed length of 4, so the prior computed for each option has its own probability distribution. For a dataset with variable lengths of option sets (3-15 options for our SATA-Bench). We can only use priors computed for their own length groups (for example, using a length-3 prior to remove bias only for questions that have 3 options). Therefore, we might not have enough data to build an accurate prior. For example, SATA-BENCH contains only 52 out of 1650 questions with 3 choices.

**Adaptation to solve SATA questions.** To solve the above problem, we first construct a dictionary with key as the lengths seen in the dataset, and value as prior computed only from questions with corresponding length, for example:
$$3: [0.5, 0.3, 0.2],$$
$$4: [0.4, 0.3, 0.1, 0.2],$$
$$N: [0.2, 0.1, 0.1, 0.04, 0.04, 0.01, ...]$$

To supplement the lengths with lower datapoint, we take prefix of the longer priors, then *normalize* to unit vector, and use as auxiliary datapoints to help computing for shorter priors, for example a 10-option prior (prior computed from 10-option question) can be used to help computing priors for 3-option question:
$$[\textbf{0.12}, \textbf{0.2}, \textbf{0.05}, 0.17, 0.04, 0.01, 0.01, 0.02, 0.3, 0.2]$$

$$\downarrow$$

$$[\textbf{0.32}, \textbf{0.54}, \textbf{0.14}]$$

We take the first 3 numbers corresponding to "ABC" of a 3-option question, then normalize it to the unit vector with the same probability distribution as the other 3-option priors. Similarly, this 10-option prior can also be used to compute priors for any shorter lengths.

Lastly, because Choice Funnel will remove the selected option from the option set, the option IDs (ABCD) would not be continuous. Because the prior vector can only work with a continuous option set, we must **rebalance the option IDs**. For example, "ACDE" ("B" is removed) will be rebalanced to "ABCD".

### M.3 Conclusion and Takeaways

Once we have done this process we should have a large enough population to compute accurate priors for most lengths. One limitation is that this adaptation does not help much if we don't have enough questions for longer lengths in our dataset, though this is not the case for SATA-Bench, which contains 21.88% data for its longest 15-option question. One potential solution is to use synthetic datasets to backfill longer-option questions, since the original work showed that the prior is transferable. We leave this for future work.

## N Experiment Setup for Choice Funnel

We chose a fixed *90%* confidence threshold as the stopping condition (ii) in Choice Funnel for **all models**. While this initial parameter selection was chosen for its simplicity, later evaluation indicated that it yielded sufficiently robust performance. Consequently, we did not pursue further investigation into more granular threshold adjustments. It also demonstrates that the algorithm is generalizable to other models without careful calibration.

The first baseline method *first token* sets a fixed threshold so that any option with a probability above the threshold is selected, and this should be the lower bound of the performance. *First token debiasing* can be used to find out if the popular strategy used to solve the MCQ questions is transferable to the SATA questions in terms of minimizing the impact of the selection bias. Lastly, we expect *yes/no* to be a competitive baseline given that it processes each choice separately.

**Prompts.** To reduce the bias introduced by prompt design and emphasize the impact of the method itself, we choose prompts for all methods with minimal engineering effort and mainly capture the essential components: *paragraph, question and choices*. The complete prompts are given in Appendix J.

**Models.** Our study focuses on the causal, decoder-only LLMs since this architecture has become the dominant choice for modern LLMs. We experiment with *7 LLMs from Table 2 under Probability Based Retrieving* which are all popular open-source models on the HuggingFace website, and we can access their output probabilities: DeepSeek R1 Distilled LLAMA 8B [DeepSeek-AI et al., 2025], Qwen2.5 14B [Yang et al., 2025], Ministral 8B [Team, 2024], Phi 3 7B [Abdin et al., 2024], Phi 4 mini reasoning [Abdin et al., 2025], Bloomz 7B [Muennighoff et al., 2022], and Llama 3.1 8B [Touvron et al., 2023].

## O Ablation Study for Choice Funnel

### O.1 "I don't know" performs worse than "None of the above"

Table 11: Performance comparison of Choice Funnel using "None of the Above" versus "I don't know" options.

| Method | EM↑ | Precision↑ | Recall↑ | JI↑ | SPD↓ | CtDifAbs↓ | CtAcc↑ | InfCost↓ |
|---|---|---|---|---|---|---|---|---|
| Phi3-7B + *nota* | **29.27** | **83.27** | 70.24 | 61.85 | 3.47 | **1.42** | **0.38** | 6339 |
| Phi3-7B + *idk* | 28.18 | 80.92 | **73.25** | **62.22** | **2.35** | 1.48 | 0.36 | 6667 |
| Llama3-8B + *nota* | **19.88** | **78.69** | 56.19 | **50.36** | 7.74 | **1.66** | **0.33** | **4975** |
| Llama3-8B + *idk* | 17.64 | 75.50 | **58.03** | 49.55 | 7.74 | 1.69 | 0.32 | 5066 |
| Bloomz-7B + *nota* | **20.18** | **66.62** | 54.90 | **46.15** | 17.78 | **1.71** | **0.32** | **5440** |
| Bloomz-7B + *idk* | 18.00 | 65.55 | **55.76** | 45.53 | **16.45** | 1.76 | 0.31 | 5528 |

We compared two commonly employed auxiliary response options in traditional survey science domain [Schuman and Presser, 1996]: 'I don't know' (*IDK*) and 'None of the above' (*NOTA*),

examining their effectiveness as *Choice Funnel* stopping condition. Based on an ablation study on Table 13, *NOTA* yields consistently better performance. When using *IDK*, we observe **noticeable increase in *InfCost* and result in worse Count Bias** (*CtDifAbs* and *CtAcc*), which means **model tends to over select number of options**, indicating that the model would rather select a wrong answer than saying "I don't know". This is potentially related to RLHF process, where the model is trained to generate answers that are more favorable to humans.

## O.2 Ablation on Choice Funnel Components

Table 12: Ablation study demonstrating that PriDe token debiasing effectively mitigates unselection bias.

| Method | EM↑ | Precision↑ | Recall↑ | JI↑ | SPD↓ | CtDifAbs↓ | CtAcc↑ | InfCost↓ |
|---|---|---|---|---|---|---|---|---|
| Phi3-7B + *debiasing only* | 1.76 | 67.92 | 28.24 | 27.47 | 175.24 | 2.50 | 0.05 | **2534** |
| Phi3-7B + *CF only* | 26.00 | 80.84 | 70.08 | 60.33 | 4.17 | 1.44 | 0.35 | 6436 |
| Phi3-7B + *CF + debiasing* | **29.27** | **83.27** | **70.24** | **61.85** | **3.47** | **1.42** | **0.38** | 6339 |
| Llama3-8B + *debiasing only* | 7.58 | 62.83 | 32.28 | 30.38 | 151.74 | 2.34 | 0.14 | **2534** |
| Llama3-8B + *CF only* | 17.45 | 76.37 | 50.84 | 46.74 | 10.12 | 1.67 | **0.34** | 4380 |
| Llama3-8B + *CF + debiasing* | **19.88** | **78.69** | **56.19** | **50.36** | **7.74** | 1.66 | 0.33 | 4975 |
| Bloomz-7B + *debiasing only* | 7.09 | 59.07 | 38.41 | 32.05 | 149.17 | 2.19 | 0.15 | **2534** |
| Bloomz-7B + *CF only* | 16.36 | 66.10 | 48.26 | 42.66 | 23.09 | **1.65** | **0.35** | 4469 |
| Bloomz-7B + *CF + debiasing* | **20.18** | **66.62** | **54.90** | **46.15** | 17.78 | 1.71 | 0.32 | 5440 |

We conducted an ablation study on the two sub-components of Choice Funnel: token debiasing (*"debiasing only"*) and iterative selection (the process of iteratively selecting options until a stopping condition is met, denoted as *"CF only"*). The analysis is performed on 3 open-source models. When comparing *"CF only"* to the complete *"CF + debiasing"*, the observed increase in SPD metric demonstrates that **token debiasing effectively mitigates unselection bias**, yielding better performance. Nevertheless, the comparison between *"debiasing only"* and *"CF only"* reveals that **our novel iterative selection component contributes more substantially to overall performance improvements.**

## O.3 Ablation on Choice Funnel Stopping Condition

Table 13: Ablation study on the two stopping conditions in Choice Funnel, showing that combining both yields the best performance.

| Method | EM↑ | Precision↑ | Recall↑ | JI↑ | SPD↓ | CtDifAbs↓ | CtAcc↑ | InfCost↓ |
|---|---|---|---|---|---|---|---|---|
| Phi3-7B + *thresholding only* | 3.82 | 65.00 | 74.84 | 48.93 | 3.37 | 2.22 | 0.13 | 7416 |
| Phi3-7B + *NOTA only* | 29.21 | 77.07 | **85.63** | **68.00** | **0.69** | **1.20** | 0.37 | 9380 |
| Phi3-7B + *thresholding + NOTA* | **29.27** | **83.27** | 70.24 | 61.85 | 3.47 | 1.42 | **0.38** | 6339 |
| Llama3-8B + *thresholding only* | 0.89 | 71.92 | 52.22 | 44.12 | 10.53 | 1.74 | 0.27 | 4564 |
| Llama3-8B + *NOTA only* | 19.51 | 69.22 | **85.77** | **60.09** | **2.24** | 1.94 | 0.25 | 10212 |
| Llama3-8B + *thresholding + NOTA* | **19.88** | **78.69** | 56.19 | 50.36 | 7.74 | **1.66** | **0.33** | 4975 |
| Bloomz-7B + *thresholding only* | 9.94 | 64.47 | 48.93 | 40.77 | 22.50 | 1.72 | 0.29 | 4506 |
| Bloomz-7B + *NOTA only* | 12.24 | 55.60 | **89.57** | **52.81** | **12.82** | 3.31 | 0.17 | 13758 |
| Bloomz-7B + *thresholding + NOTA* | **20.18** | **66.62** | 54.90 | 46.15 | 17.78 | **1.71** | **0.32** | 5440 |

We conducted an ablation study to evaluate the relative importance of our two proposed stopping conditions in Choice Funnel. The results demonstrate that Choice Funnel achieves optimal performance when both conditions are applied in combination. Notably, the "None of the above" (*NOTA*) condition emerged as the more influential factor, suggesting that models can reliably identify when no correct answers remain among the provided options.

# P Positional Bias Under Randomized Answer Orderings

**Does the benchmark include randomized answer orderings?** No. In the main benchmark, each question's answer choices appear in a fixed, canonical order. To quantify the extent to which large language models (LLMs) rely on this implicit positional cue, we ran an auxiliary study in which the answer choices for every question were *randomly permuted* (e.g. A B C → C A B). We then compared model performance on the permuted dataset to its performance on the original version.

**Setup.** All hyper-parameters, prompts, and decoding settings were kept *identical* to the main benchmark; only the answer order was shuffled once per question. Table 14 reports the *difference* (*permute–original*) for each metric, so negative values indicate a drop in performance and positive values indicate an increase. [†] **CtDif** is shown with a downward arrow even though its baseline values are negative; a more negative CtDif therefore indicates a larger absolute mismatch in option counts.

Table 14: Change in evaluation metrics after randomly reordering answer choices. Performance metrics are expected to **increase** (↑) while bias metrics are expected to **decrease** (↓).

| Model | EM ↑ | Precision ↑ | Recall ↑ | JI ↑ | RStd ↓ | RSD ↓ | SPD ↓ | CtDif[†] ↓ | CtDifAbs ↓ |
|---|---|---|---|---|---|---|---|---|---|
| Claude 3 Haiku | −24.06 | −34.69 | −34.28 | −35.31 | +6.06 | +0.17 | +0.12 | −0.07 | −0.51 |
| Llama 3.1 405B | −3.80 | −3.90 | −4.71 | −5.22 | +9.73 | −0.20 | +0.25 | −0.18 | −0.71 |

**Findings.** All three models suffer performance degradation when answer choices are shuffled, with **Claude 3 Haiku** exhibiting the sharpest decline (–24 EM, –35 JI). Selection / count-bias metrics (RStd, SPD, CtDifAbs) *increase* for every model except RSD, confirming heightened positional bias.

**Discussion.** These results suggest that current LLMs implicitly learn positional heuristics from training data in which answer orders are fixed. Breaking this assumption makes the models less certain and more prone to biased guessing. Future work should examine (i) whether fine-tuning on randomly ordered choices mitigates the effect, and (ii) how pronounced the bias is for other model families and task domains.

## Q  Per-Dataset Performance Breakdown

We report detailed bias metrics for different task categories in Table 15. The News dataset has the lowest selection bias, while Reading Comprehension exhibits the highest. For count bias, Toxicity shows the smallest difference, and Biomedicine has the largest. Notably, News has significantly lower selection and count biases compared to other datasets (p-values: 0.03 for SPD and $3.8 \times 10^{-5}$ CtDifAbs, T-test). All datasets show negative count difference, confirming underprediction and the presence of count bias in SATA questions.

Table 15: Breakdown of Bias metrics by subject. Lower values are better for all metrics.

| Task | RStd ↓ | RSD ↓ | SPD ↓ | CtDif | CtDifAbs ↓ |
|---|---|---|---|---|---|
| Reading Comprehension | 19.29 ± 7.59 | 0.20 ± 0.10 | 1.53 ± 1.39 | -0.68 ± 0.42 | 0.85 ± 0.35 |
| Toxicity | 7.13 ± 2.83 | 0.11 ± 0.07 | 0.48 ± 0.56 | -0.05 ± 0.44 | 1.28 ± 0.16 |
| News | 4.32 ± 3.16 | 0.08 ± 0.19 | 0.12 ± 0.23 | -0.09 ± 0.25 | 0.32 ± 0.19 |
| Biomedicine | 6.66 ± 2.37 | 0.15 ± 0.14 | 2.90 ± 3.60 | -1.71 ± 0.96 | 2.22 ± 0.67 |
| Laws | 5.75 ± 4.17 | 0.13 ± 0.16 | 1.54 ± 3.43 | -1.00 ± 0.87 | 1.36 ± 0.75 |
| Events | 7.15 ± 4.14 | 0.13 ± 0.19 | 0.85 ± 1.02 | -0.28 ± 0.77 | 1.08 ± 0.30 |

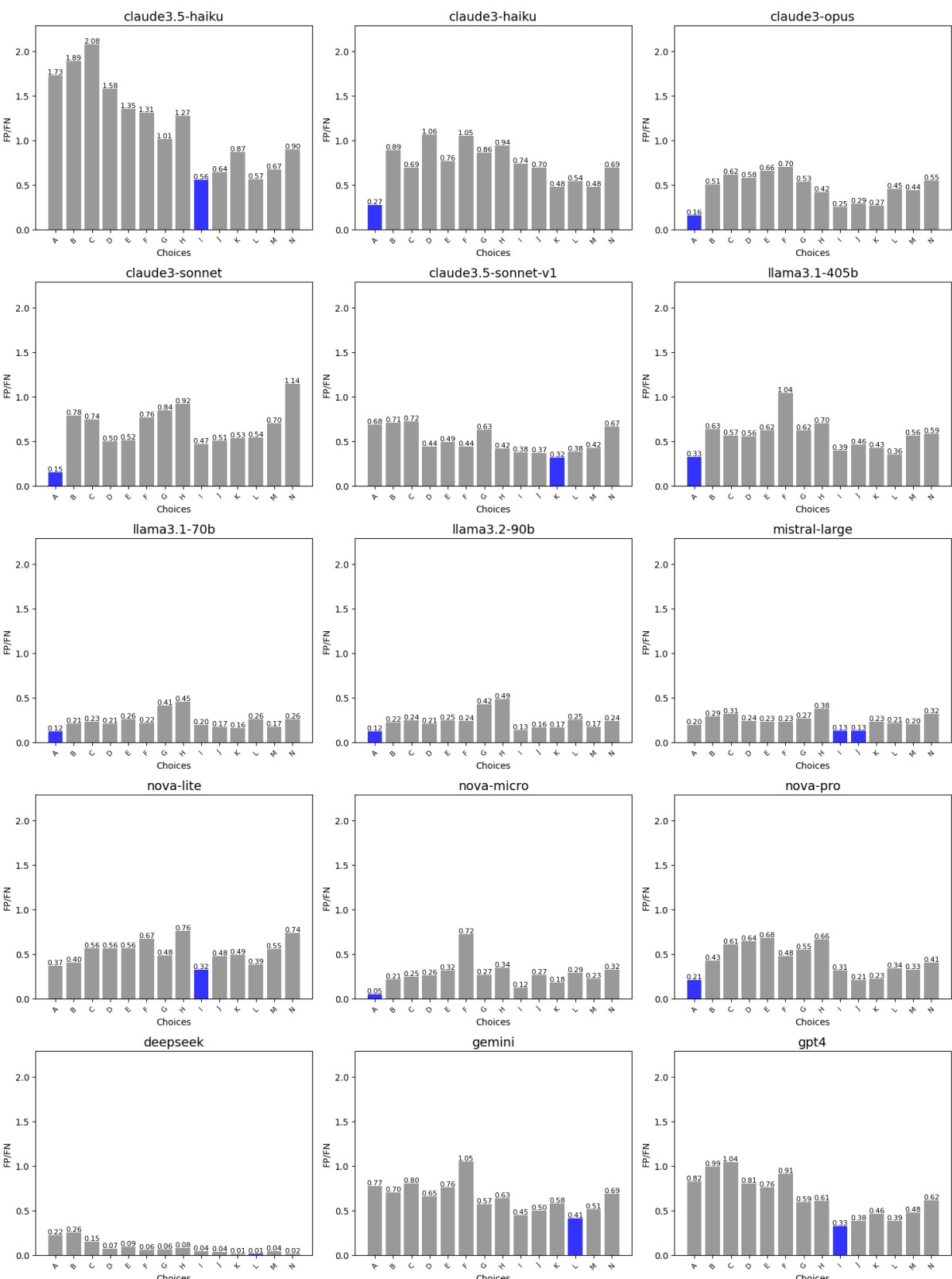

Figure 7: Ratio of false positive rate to false negative rate per label for each evaluated LLM.

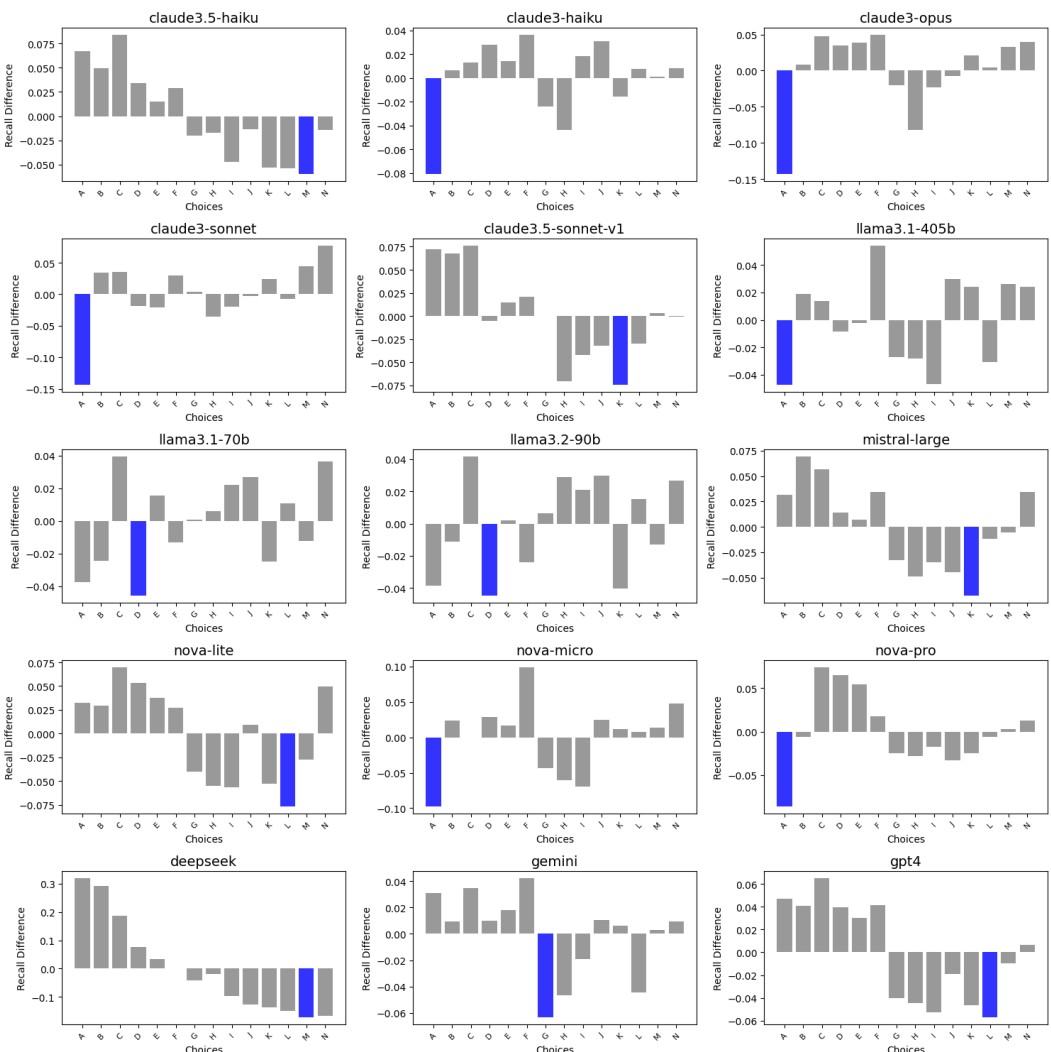

Figure 8: Recall score per label (Y-axis), normalized by subtracting the model's average recall. Most models exhibit at least one label with significantly lower recall than the rest.

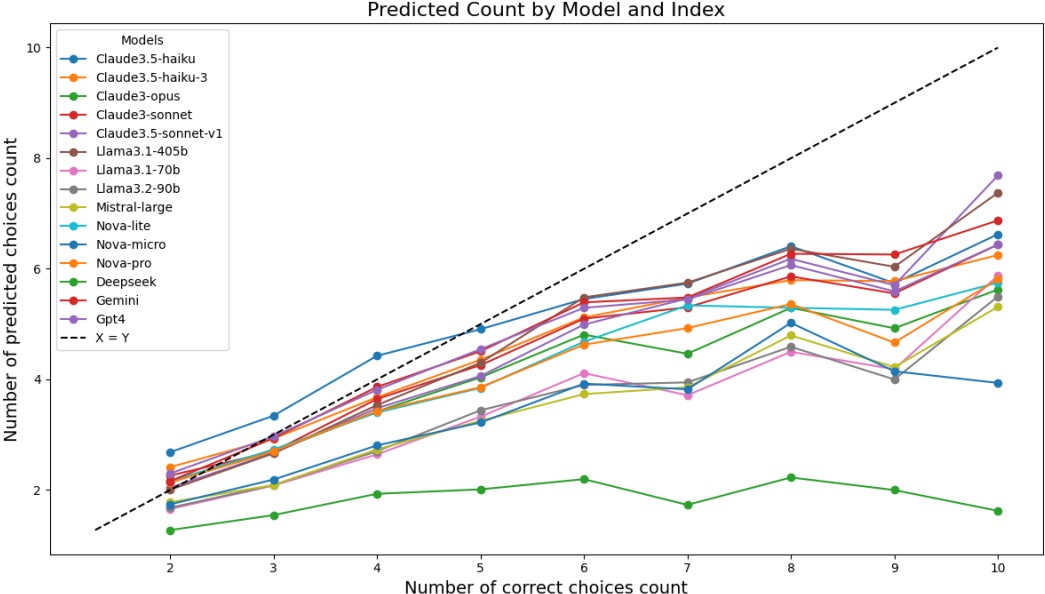

Figure 9: Relationship between predicted and actual correct choice counts across models. Models generally under-select the correct number of answer choices. Y-axis represents the average number of choices selected by the model. X-axis represents the actual number of correct choices. A perfect model would align along the diagonal where X equals Y.

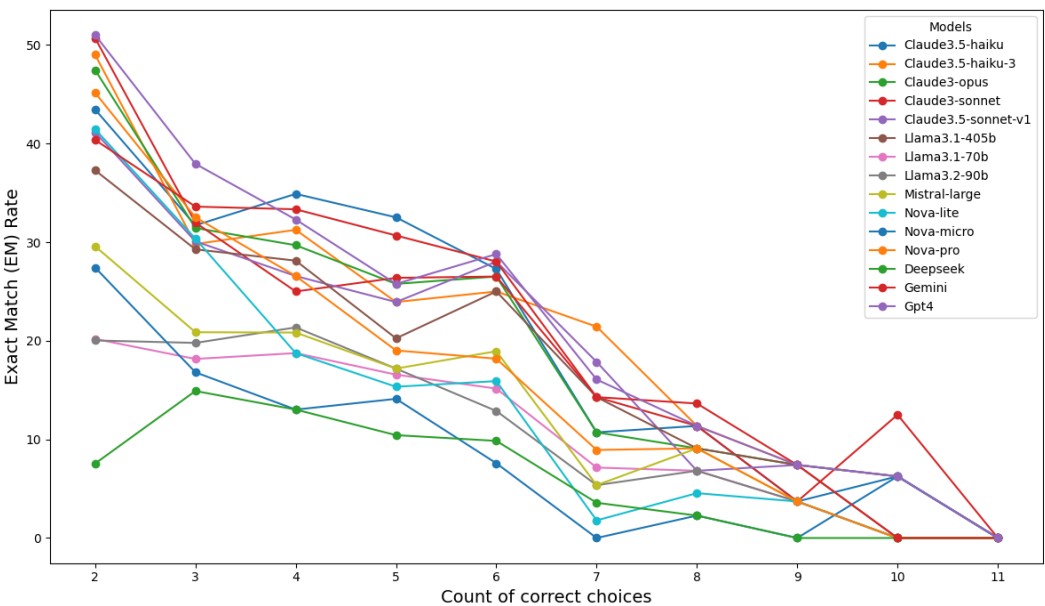

Figure 10: Relationship between Exact Match Rate and the number of correct choices. As the number of correct choices increases, the exact match rate decreases. None of the models achieve an exact match rate above 20% when the number of correct choices exceeds 7.

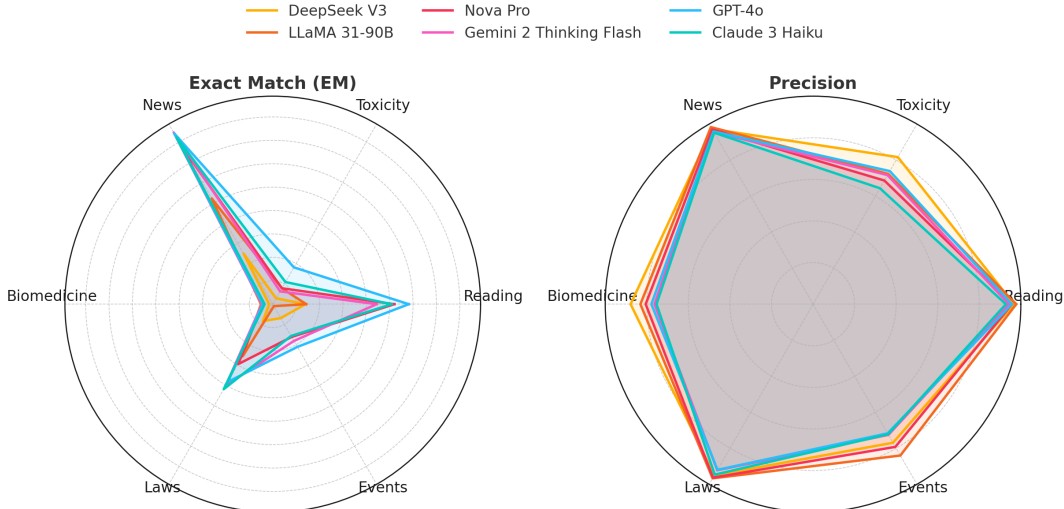

Figure 11: Performance breakdown of evaluated models across different source datasets.

