# OpenReview forum: "SATA-BENCH: SELECT ALL THAT APPLY BENCH- MARK FOR MULTIPLE CHOICE QUESTIONS"
_EurIPS.cc/2025/Workshop/UPLB — UPLB2025_

### Official Review · Reviewer_BifP · 2025-10-27
**Review of 'SATA-Bench: Select All That Apply Benchmark for Multiple Choice Questions'**

**Rating:** 8
**Confidence:** 3

**Review:**

The paper investigates the ability of current Large Language Models (LLMs) to identify the correct answers in multiple choice questions, i.e. when the number of correct answers is at least two. This setting applies to several real-world problems where multiple categories have to be assigned to a given object.

To address this task, the authors build a benchmark datasets of 10000 human-validated questions spanning six domains: reading comprehension, toxicity detection, news categorization, biomedicine, legal classification, event analysis.
The authors show that current models, including both proprietary and open source, struggle with assigning the correct labels to a given text. Overall, among the 32 LLMs analyzed, the maximal fraction of exact matches was around 42 %.
They analyze the source of this error and spot three sources of bias: 'unselection' (models avoid certain labels systematically, regardless of the content and correctness of the option), 'speculation' (models provide incorrect answers when uncertain) and 'count' (models tend to underestimate the number of correct answers). To tackle this biases, the authors explore different mitigation strategies, proposing an algorithm (Choice Funnel) to improve models performances against unselection and count biases.

The analysis performed by the authors is robust enough for being accepted to the UPLB workshop.
The biases highlighted by the authors are likely a consequence of the training procedure of the models, mostly based on binary decision tasks (yes/no answers or single correct answer among multiple options), even though I wonder if some aspects of these biases may be intrinsic to models; for instance, does the average count difference depend on the number of correct options? It may be interesting to see how LLMs behave with respect to questions with 'few' correct options versus questions with 'many' correct options.

To conclude, I believe the paper content is interesting and can trigger interesting discussions on site during the workshop. Indeed, given the widespread use of LLMs and their societal impact, understanding failure modes and biases in existing models is of great value for science and applications.

---

### Decision · Program_Chairs · 2025-11-03

Accept (Poster)